# Identification of serum metabolites associating with chronic kidney disease progression and anti-fibrotic effect of 5-methoxytryptophan

Dan-Qian Chen [1], Gang Cao[2], Hua Chen[1], Christos P. Argyopoulos[3], Hui Yu[3], Wei Su[4], Lin Chen[1], David C. Samuels[5,6], Shougang Zhuang[7,8], George P. Bayliss[8], Shilin Zhao[5], Xiao-Yong Yu[9], Nosratola D. Vaziri[10], Ming Wang[1], Dan Liu[1], Jia-Rong Mao[9], Shi-Xing Ma[4], Jin Zhao[11], Yuan Zhang[11], You-Quan Shang[4], Huining Kang[3], Fei Ye[5], Xiao-Hong Cheng[9], Xiang-Ri Li[12], Li Zhang[11], Mei-Xia Meng[11], Yan Guo[1,3] & Ying-Yong Zhao [1]

Early detection and accurate monitoring of chronic kidney disease (CKD) could improve care and retard progression to end-stage renal disease. Here, using untargeted metabolomics in 2155 participants including patients with stage 1–5 CKD and healthy controls, we identify five metabolites, including 5-methoxytryptophan (5-MTP), whose levels strongly correlate with clinical markers of kidney disease. 5-MTP levels decrease with progression of CKD, and in mouse kidneys after unilateral ureteral obstruction (UUO). Treatment with 5-MTP ameliorates renal interstitial fibrosis, inhibits IκB/NF-κB signaling, and enhances Keap1/Nrf2 signaling in mice with UUO or ischemia/reperfusion injury, as well as in cultured human kidney cells. Overexpression of tryptophan hydroxylase-1 (TPH-1), an enzyme involved in 5-MTP synthesis, reduces renal injury by attenuating renal inflammation and fibrosis, whereas TPH-1 deficiency exacerbates renal injury and fibrosis by activating NF-κB and inhibiting Nrf2 pathways. Together, our results suggest that TPH-1 may serve as a target in the treatment of CKD.

[1] Faculty of Life Science & Medicine, Northwest University, No. 229 Taibai North Road, Xi'an, Shaanxi 710069, China. [2] School of Pharmacy, Zhejiang Chinese Medical University, No. 548 Binwen Road, Hangzhou, Zhejiang 310053, China. [3] Department of Internal Medicine, University of New Mexico, 1700 Lomas Blvd NE, Albuquerque, New Mexico 87131, USA. [4] Department of Nephrology, Baoji Central Hospital, No. 8 Jiangtan Road, Baoji, Shaanxi 721008, China. [5] Department of Biomedical Informatics, Vanderbilt University Medical Center, 1211 Medical Center Dr, Nashville, Tennessee 37232, USA. [6] Department of Molecular Physiology and Biophysics, Vanderbilt University, 1211 Medical Center Dr, Nashville, Tennessee 37232, USA. [7] Department of Nephrology, Shanghai East Hospital, Tongji University School of Medicine, No. 150 Jimo Road, Shanghai 200120, China. [8] Department of Medicine, Rhode Island Hospital and Alpert Medical School, Brown University, 593 Eddy St, Providence, Rhode Island 02903, USA. [9] Department of Nephrology, Affiliated Hospital of Shaanxi Institute of Traditional Chinese Medicine, No. 2 Xihuamen, Xi'an, Shaanxi 710003, China. [10] Division of Nephrology and Hypertension, School of Medicine, University of California Irvine, 1001 Health Sciences Rd, Irvine, California 92897, USA. [11] Department of Nephrology, Xi'an No. 4 Hospital, No. 21 Jiefang Road, Xi'an 710004, China. [12] School of Chinese Materia Medica, Beijing University of Chinese Medicine, No. 11 North Third Ring Road, Beijing 100029, China. These authors contributed equally: Dan-Qian Chen, Gang Cao, Hua Chen. Correspondence and requests for materials should be addressed to Y.G. (email: YaGuo@salud.unm.edu) or to Y.-Y.Z. (email: zyy@nwu.edu.cn)

Chronic kidney disease (CKD) results in gradual loss of kidney function leading to end-stage renal disease, requiring dialysis or renal transplantation. While the early stages of CKD, stages 1–2, have little signs or symptoms and the disease is often not detected until the later stages[1], the risk of cardiovascular morbidity and mortality increases with progression of CKD.

The routine clinical assessment of CKD is based on measuring a marker of kidney damage (proteinuria) and estimate renal function (estimated Glomerular Filtration Rate, eGFR). However estimation of eGFR using the serum creatinine may be affected by non-renal processes (e.g., muscle mass or advanced age) and as such is associated with substantial error[2]. At the same time, assessment of proteinuria (albuminuria) may also be affected by non-renal processes (e.g. fever, infection). Furthermore, proteinuria may persist long after the successful resolution of kidney injury in many primary renal diseases leading to unnecessary escalation of treatment for many patients[3,4]. Since the potential for mislabeling CKD[5] status on the basis of eGFR and the assessment of kidney damage on the basis of proteinuria is very high, there is an urgent need for better assessments of renal function to improve the accuracy of the diagnosis of CKD.

Use of high-throughput and high-resolution metabolomics analyses of human plasma in large-scale epidemiologic studies has enabled metabolic phenotyping in well-characterized human cohorts[6–10]. Metabolomics has the potential to provide insight into the disease mechanisms by illuminating the underlying metabolic pathways and to discover potential biomarkers[11]. We believe that metabolomics hold the potential to improve diagnosis and management of CKD by providing stage-specific and injury detecting biomarkers.

Several recent reviews have reported instances of metabolomics applied in the identification of biomarkers for renal diseases[12–15]. This appears to be an exciting opportunity for this class of biomarkers relative to multivariate proteomic markers, which classify patients at high risk of progression in early CKD[16,17]. Clinical risk assessment of eGFR and albuminuria is inefficient in patients with early CKD, impeding the development of alternative therapies[18] or even targeting patients at risk for early intervention to prevent the development of kidney damage[19]. We anticipate that our identified metabolites will improve on the existing clinical markers and meet two large and related unmet needs in the areas of drug development and clinical practice.

In this study we enroll 2155 subjects to identify potential early stage CKD metabolite biomarkers using ultra-performance liquid chromatography coupled with quadrupole time-of-flight SYNAPT high-definition mass spectrometry (UPLC-HDMS) which has been shown to have excellent selectivity, sensitivity and reproducibility[20–24]. The anti-inflammatory and anti-fibrotic effects of the most promising metabolite 5-methoxytryptophan (5-MTP) and the biological roles of its regulatory enzyme tryptophan hydroxylase-1 (TPH-1) are examined in cell and animal models. The present study demonstrates five metabolites can separate patients with early stage CKD from healthy controls and serve as accurate biomarkers of the early stage of CKD. Mechanistically, the loss of 5-MTP and its regulatory enzyme, TPH-1, contribute to kidney injury by activating NF-κB and inhibiting nuclear factor-erythroid-2-related factor 2 (Nrf2) pathways. Therefore, TPH-1 may serve as a target in the treatment of CKD.

## Results

### Demographic characteristics of the study population.
Flow diagram of the overview of study design was depicted in Supplementary Fig. 1. The general clinical and demographic data on

healthy controls and patients with stage 1–5 CKD for the training data set are presented in Supplementary Table 1. Age, body weight (BW), body mass index (BMI), blood pressure, eGFR, serum creatinine (CREA) and urea concentrations, proteinuria and urine protein/creatinine ratio were significantly associated with CKD stages.

The independent validation dataset contains 30 healthy controls, 30 patients with CKD1 and 30 patients with CKD2. CREA levels of each group were $64.4 \pm 14.8$, $66.8 \pm 17.6$ and $77.2 \pm 20.5$ μmol/l, respectively. Similarly, the urea level of healthy controls, CKD1 and CKD2 were $5.3 \pm 1.4$, $5.2 \pm 1.5$ and $6.1 \pm 2.4$ mmol/l, respectively. These results demonstrated that serum CREA and urea were not significantly different among the three groups.

In order to further validate the biomarkers, we used data from a longitudinal cohort study composed of 1248 participants with a normal eGFR ($91.4 \pm 17.0$ ml/min/1.73 m$^2$) at baseline. In that cohort 31 participants had a decrease of eGFR ($52.5 \pm 6.6$ ml/min/ 1.73 m$^2$) and they developed CKD during the six-year follow-up period. Potential metabolite biomarkers were also validated by drug intervention on 114 patients with CKD2.

### Metabolomic analysis and five metabolite identification.
Using UPLC-HDMS, we detected 25109 variables including 16382 in positive ion mode and 8727 in negative ion mode from 703 subjects in our initial dataset. Based on least absolute shrinkage and selection operator (LASSO)-based variable selection, we identified 58 and 66 variables in the training datasets from positive and negative ion modes, respectively. Of these 124 variables, 98 metabolites including 39 in positive ion mode and 59 in negative ion mode were identified based on MS data, MS$^E$ fragmentation, elemental compositions and available databases (Supplementary Data 1). The metabolite identification methods were presented in Methods section. A multivariate ordinal regression model was fit by using all 39 metabolites identified from the positive ion mode to determine their effect size (CKD ~ metabolites). McFadden's pseudo R$^2$ were computed for the positive ion mode at 95.2%, suggesting that 95.2% of the variation in the CKD stage can be explained by these 39 metabolites. The McFadden's pseudo R$^2$ for the 59 metabolites in negative ion mode was 94.9%. A univariate ordinal regression model was fit using these 98 metabolites (Supplementary Data 1).

To further test the classification power of these 98 metabolites, we performed random forest and support vector machine classification, and achieved an accuracy of 98.4% for random forest method, and 99.0% for support vector machine method. Metabolic pathway analysis indicated that 98 metabolites involved in Farnesoid X receptor/Retinoid X receptor (FXR/ RXR) activation, aspartate biosynthesis and degradation, glutamate degradation, acetyl-CoA biosynthesis, spermine and spermidine degradation, guanosine nucleotides degradation, serotonin and melatonin biosynthesis and hypusine biosynthesis (Supplementary Fig. 2).

To further reduce the number of metabolites for clinical application and functional analysis purpose, we conducted one more round of feature reduction on these 98 metabolites plus the 17 clinical indexes. This model selected the five metabolites presented in Supplementary Table 2. Five metabolites including 5-MTP, canavaninosuccinate (CSA), acetylcarnitine, tiglylcarnitine and taurine can clearly distinguish all clinical stages of CKD (Fig. 1a–e and Supplementary Table 2). The ordinal regression LASSO model also revealed that CREA and urea that can clearly distinguish late stage CKD (Fig. 1f, g). However they are not effective at separating patients with CKD1 from healthy controls. Proteinuria also demonstrated large difference between patients

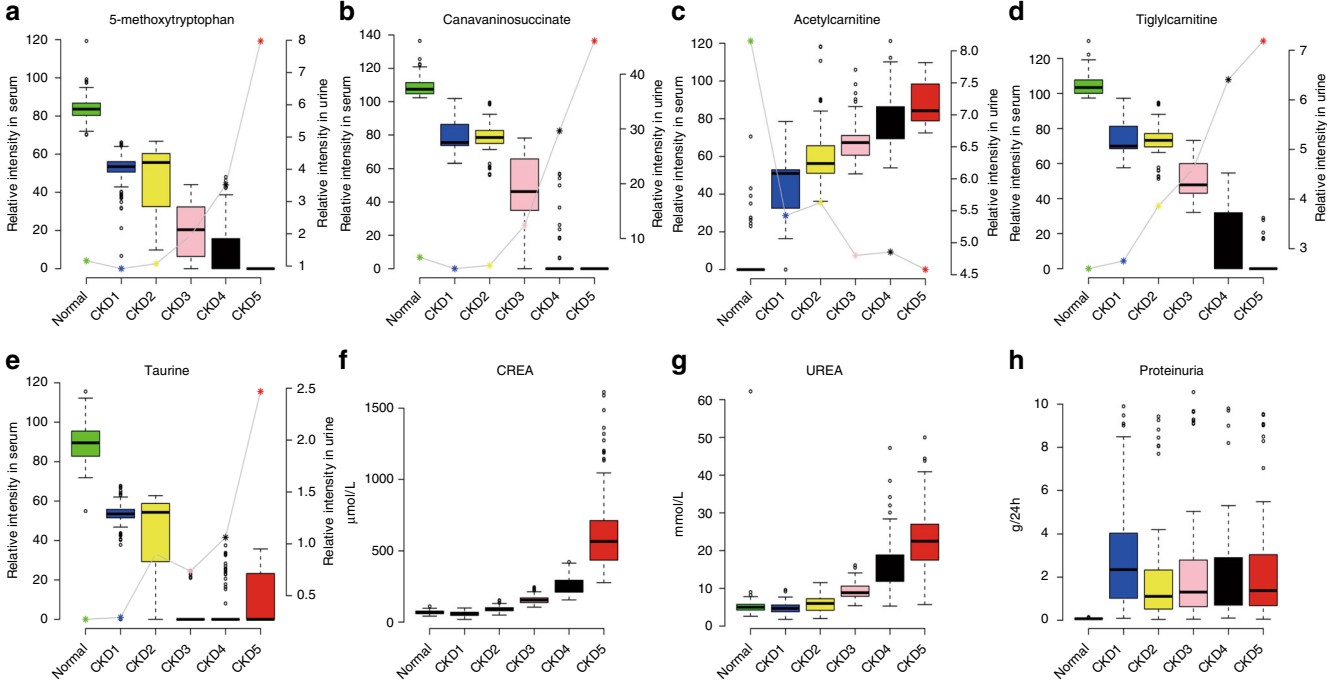

**Fig. 1** The five selected metabolites for the discrimination of the different CKD stages. **a–e** Boxplots showed the relative intensities of the five metabolites in serum (left y-axis) and urine (right y-axis) including 5-MTP, CSA, acetylcarnitine, tiglylcarnitine and taurine across all five CKD stages. **f, g** The levels of serum CREA and urea across all five CKD stages. **h** The levels of proteinuria across all five CKD stages. Dot plot and line showed the relative intensities of the five urinary metabolites including 5-MTP, CSA, acetylcarnitine, tiglylcarnitine and taurine across all five CKD stages. In the boxplot, the median is represented by the center line, 75 percentile is represented by the upper bound of the box, 25 percentile is represented by the lower bound of the box, minimum is represented by the lower whisker, and maximum is represented by the upper whisker

with CKD and healthy controls (Fig. 1h). Four metabolites had negative associations (5-MTP $p = 8.22 \times 10^{-13}$, CSA $p = 2.99 \times 10^{-5}$, tiglylcarnitine $p = 7.17 \times 10^{-9}$ and taurine $p = 1.63 \times 10^{-7}$), and acetylcarnitine had a positive association ($p = 3.19 \times 10^{-13}$). The changes of five metabolites are displayed in Fig. 1a–e.

The final model with the five metabolites explained 94.1% of the variation in different CKD stages. Unsupervised clustering analysis showed that five metabolites could distinguish health controls and the different stages of CKD (Fig. 2a). Supervised random forest classification using the five metabolites had a classification accuracy of 95.5%, and supervised Support Vector Machine classification showed a 92.0% accuracy (Fig. 2b–d). The receiver operating characteristic (ROC) curve showed that area under the curve (AUC) values of the five metabolites and proteinuria were equal at 0.99 (Fig. 2e), indicating that the five metabolites performed equally well as proteinuria at separating healthy controls from patients with CKD.

**External validation of five metabolites.** External validation of the five metabolites was conducted on an independent cohort with 30 healthy controls, 30 patients with CKD1 and 30 patients with CKD2. Supplementary Table 3 presents the general clinical and demographic data for this cohort. Supplementary Table 4 presents the associations between clinical variables and eGFR, all directions of associations consistent with initial findings. CKD 3–5 were not selected for external validation because our goal was focused on early stage CKD detection. Validation using unsupervised clustering, principal component analysis (PCA), supervised random forest and support vector machine revealed near perfect classification for subjects in normal healthy controls, CKD1 and CKD2 (Fig. 2f–i). The external validation results provide further evidence that five metabolites are potential

biomarkers in additional to proteinuria in separating subjects with early CKD from normal healthy controls.

**Longitudinal cohort validation of five metabolites.** We further carried out a prospective incident CKD study. Among all 1248 eligible participants with an eGFR ≥ 60 ml/min per 1.73 m² at baseline, 31 participants developed new-onset CKD (eGFR < 60 ml/min/1.73 m²) during the course of the longitudinal study. Baseline characteristics of the longitudinal cohort sample are shown in Supplementary Table 5.

The abundance of the five metabolites was measured by UPLC-MS/MS-based targeted metabolomics. Figure 3a, b present the abundance levels of the five metabolites for the 31 subjects before and after developing CKD. The direction of change for the five metabolites agreed with the directions observed in the training dataset. PCA, orthogonal partial least squares-discriminant analysis (OPLS-DA), and unsupervised clustering showed that the five metabolites can distinguish the 31 patients with CKD by their pre and post diagnosis status (Fig. 3c–e). Sensitivity and specificity of five metabolites only were 87.1 and 96.8%, respectively (Fig. 3f). Additionally, ROC analyses showed that the five metabolites have a high AUC values (Fig. 3g). Correlation between the five metabolites and eGFR (after diagnosis) was analyzed. 5-MTP, CSA, acetylcarnitine, tiglylcarnitine and taurine had a correlation coefficient of 0.91, 0.71, 0.78, 0.72 and 0.72, respectively.

**Validation of drug treatment.** Next, we examined how pharmacological therapies with enalapril, prednisone and Wulingsan (also called Oryeongsan) affected the five metabolites' abundance levels in a cohort of 114 patients with CKD2. The eGFR value was not significantly different between pre-treatment (68.7 ± 13.4 ml/

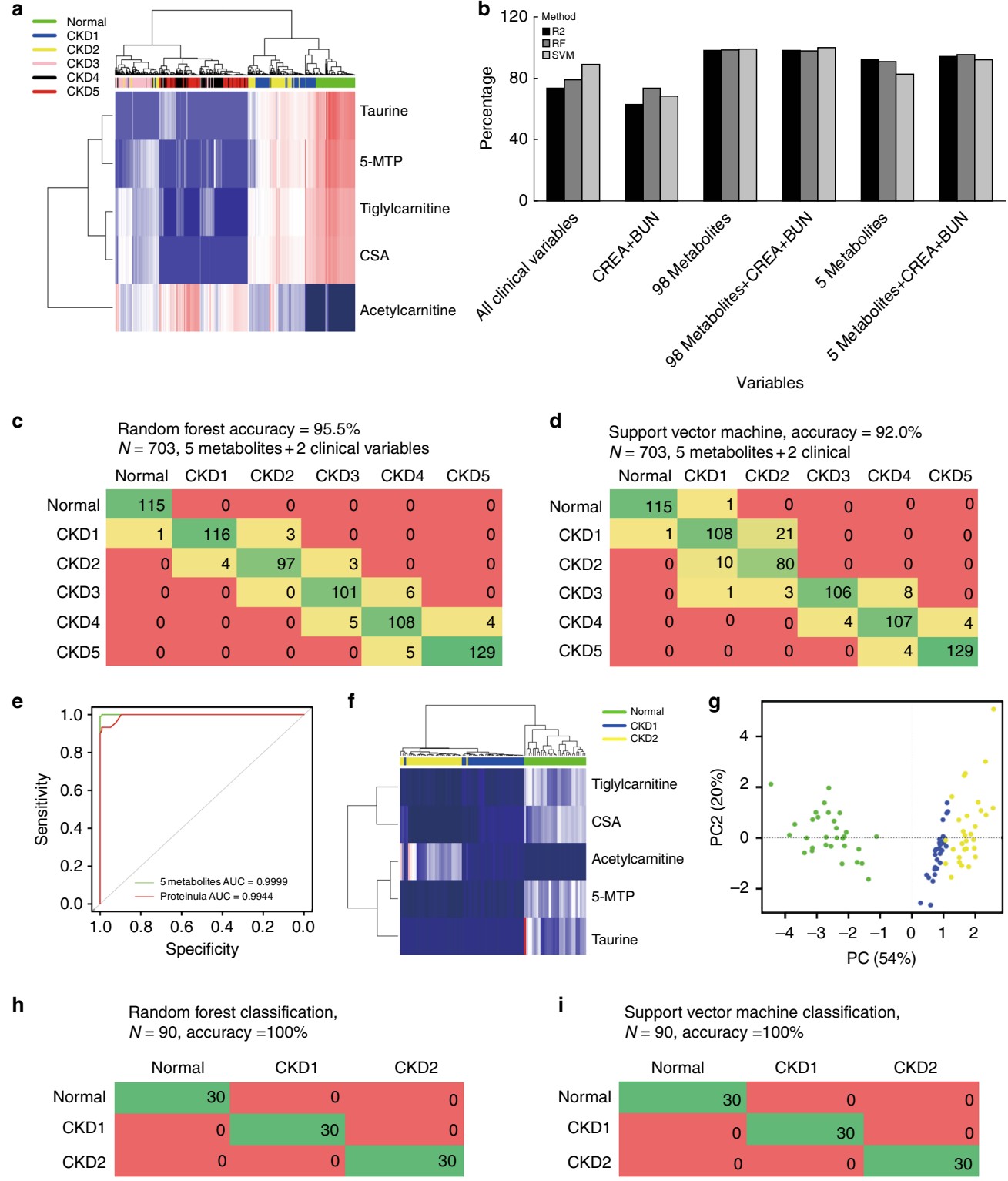

min/1.73 m$^2$) and post-treatment (71.4 ± 15.4 ml/min/1.73 m$^2$) by enalapril treatment. However, proteinuria was significantly improved after the treatment with enalapril (Fig. 4a, b), indicating therapeutic benefit of enalapril treatment. Except for taurine, the other four potential metabolites were significantly restored after patients received an enalapril treatment. Although eGFR did not change significantly during the 6-month treatment study, the five metabolites were significantly altered, which indicated the higher sensitivity of five metabolites than eGFR in early stage of CKD.

All five metabolites exhibited abundance level shifts in directions concordant with results from training dataset (Fig. 4a, b). PCA, OPLS-DA, and heatmap showed that post-treatment patients with CKD2 could be separated from pre-treatment patients with CKD2 by the combination of proteinuria and the five metabolites (Fig. 4c–e) with sensitivity and specificity of being 90.0 and 80.0%, respectively (Fig. 4f). Similar results were also observed by using only the five metabolites (Fig. 4g–j). ROC analyses showed that all five metabolites and proteinuria had high AUC (Fig. 4k).

**Fig. 2** Multivariate statistical analysis of the metabolites and clinical indexes and external validation. **a** Heatmap and unsupervised cluster constructed using the five metabolites and two clinical parameters. Strong cluster separation can be observed among CKD stages. All normal healthy controls were clustered together. Subjects with early CKD stages clearly separated from subjects with late CKD stages. Darker color was low abundance, light color was high abundance. **b** Bar plot demonstrates the accuracy and fitness of the all models tested. Three evaluation parameters were plotted: Pseudo $R^2$, random forest accuracy and support vector machine accuracy. The models with best fitness and accuracy are the models with the 98 metabolites. We further chose the model of five metabolites with two clinical parameters as our final model. **c** Confusion matrix of random forest evaluation on the final model. The accuracy was 95.5%. **d** Confusion matrix of support vector machine evaluation on the final model. The accuracy was 92.0%. **e** ROC curves of five metabolites and proteinuria in normal healthy controls and all patients with CKD. **f** Unsupervised cluster results show all 30 normal healthy controls clustered together and separated from patients with CKD. Two outliers were observed between CKD1 and CKD2. Darker color was low abundance, while light color was high abundance. **g** Scatter plot of PC1 vs PC2, clear separation can be observed between normal healthy controls and patients with CKD. **h** Random forest method was able to classify subjects to correct CKD stages. **i** Support vector machine method was able to classify subjects to correct CKD stages. AUC area under the curve, RF random forest, SVM support vector machine

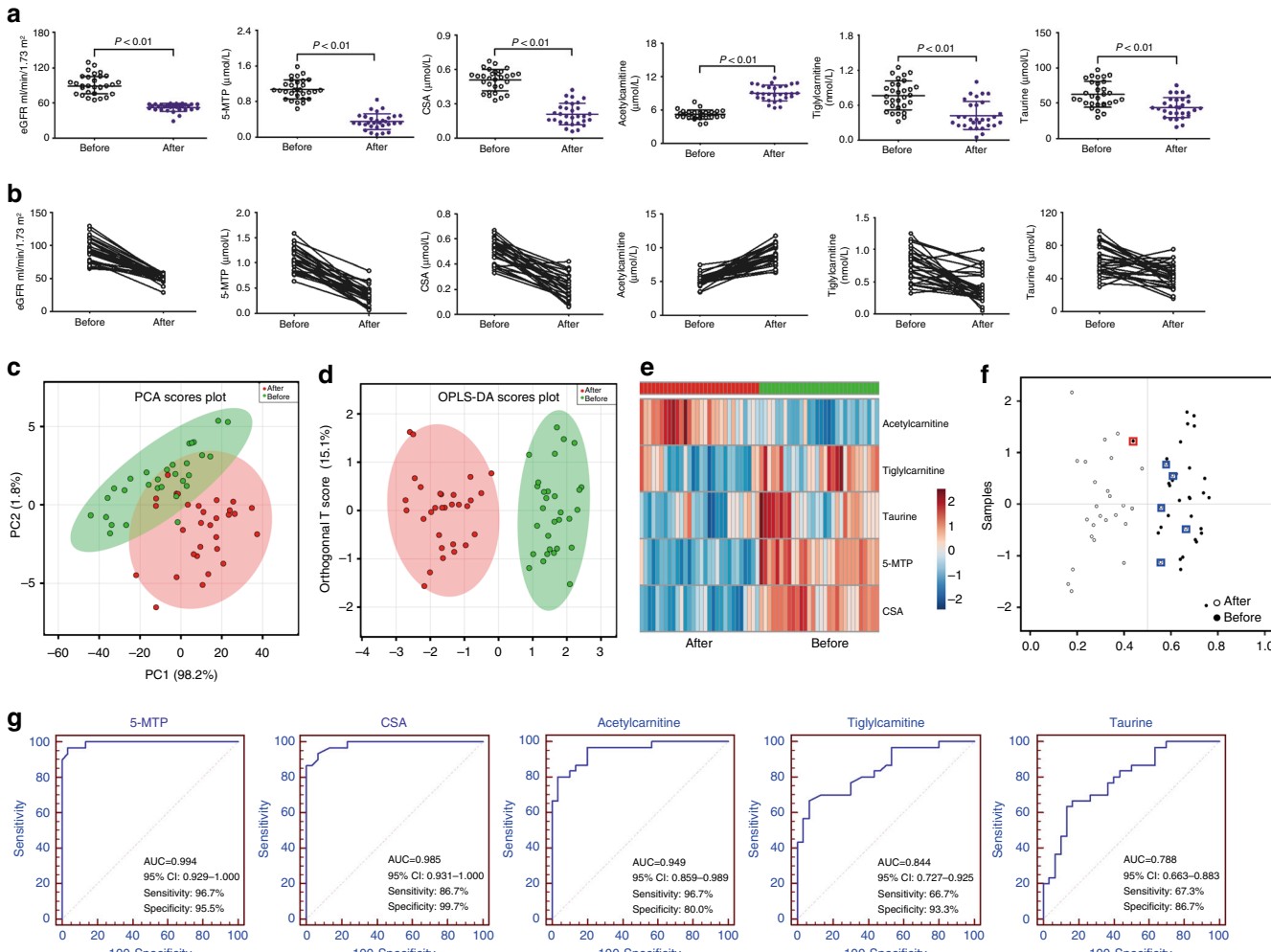

**Fig. 3** The five metabolites were further validated by a longitudinal cohort study. **a** Dot plots of levels of five metabolites including 5-MTP, CSA, acetylcarnitine, tiglycarnitine and taurine in serum of normal healthy controls and CKD case. They were determined by UPLC-HDMS method. Mean values are presented by horizontal bars. Upper and lower lines indicate standard deviation values. **b** Dots and lines showing changes of five metabolites in each individual from normal status to CKD case status. **c** PCA of two components of five metabolites from normal healthy controls and CKD case. **d** OPLS-DA of five metabolites from normal healthy controls and CKD case. **e** Heatmap of five metabolites from normal healthy controls and CKD case. **f** Diagnostic performances of five metabolites from normal healthy controls and CKD case based on the PLS-DA model. The black dots with red squares or black circles with blue squares are for the incorrectly predicted samples in CKD case and normal controls. 27 Out of the 31 CKD case were located in CKD case area (87.1% sensitivity) and 30 out of the 31 normal healthy controls were correctly grouped (96.8% specificity). These results demonstrated that the five metabolites show high prediction class probabilities. **g** Analysis of PLS-DA based ROC curves of five metabolites in normal healthy controls and CKD case from 31 individuals. The associated AUC, 95% confidence interval (CI), sensitivity and specificity values were indicated. Student's $t$ test was used for the significance of difference between two groups. OPLS-DA orthogonal partial least squares-discriminant analysis, PCA principal component analysis

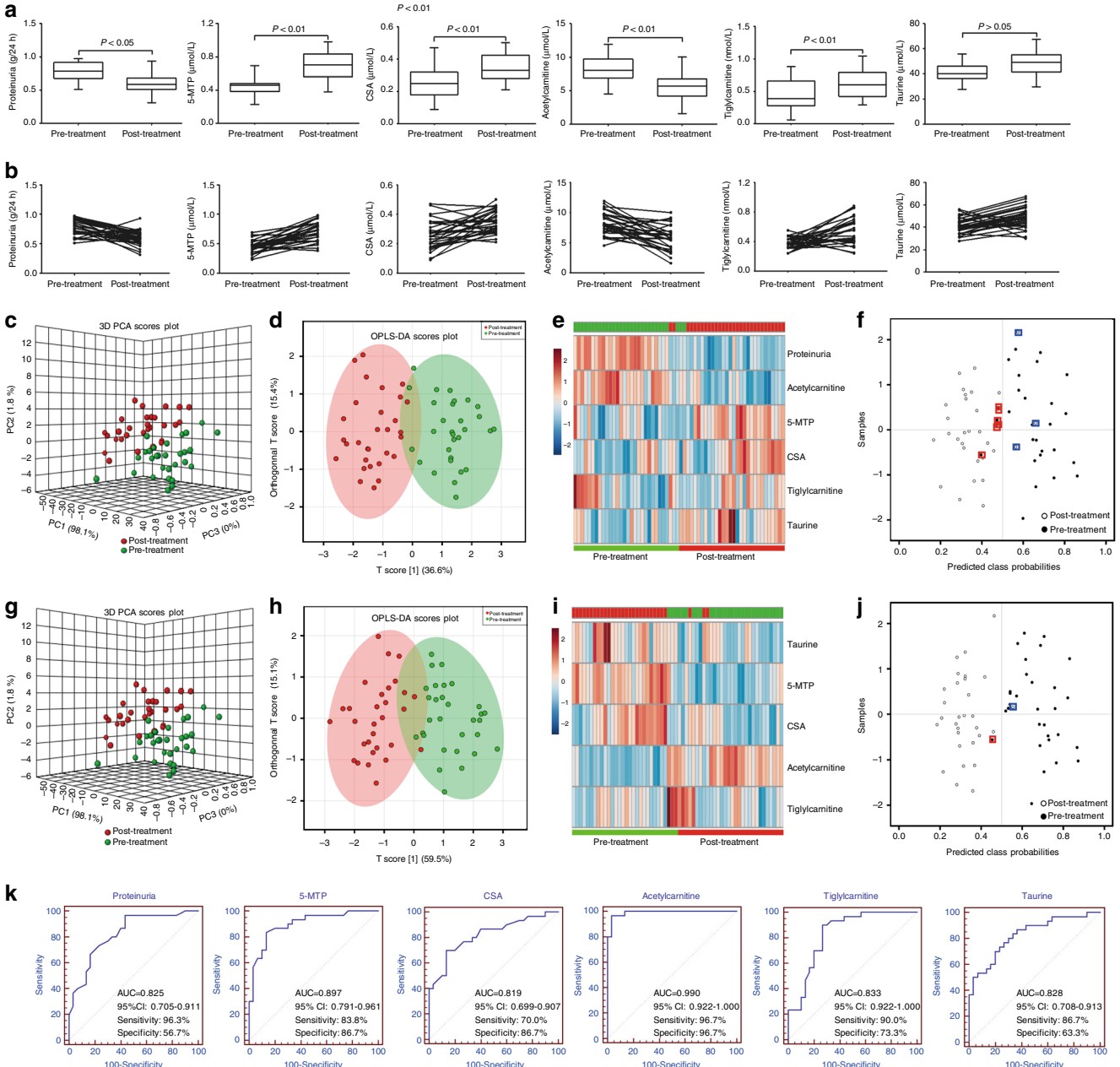

**Fig. 4** Five metabolites were further validated in pre- and post-treatment by enalapril. **a** Box plots of levels of five metabolites including 5-MTP, CSA, acetylcarnitine, tiglylcarnitine and taurine in patients with CKD2 pre- and post-treatment by enalapril. Mean values are presented by horizontal bars. The whiskers indicate the maximum and minimum points. **b** Dots and lines showing changes of five metabolites in each individual from patients with CKD2 pre- and post-treatment by enalapril. **c** PCA of two components of proteinuria and five biomarkers. **d** OPLS-DA of two components of proteinuria and five biomarkers from patients with CKD2 pre- and post-treatment by enalapril. **e** Heatmap of two components of proteinuria and five biomarkers. **f** Diagnostic performances of two components of proteinuria and five biomarkers based on the PLS-DA model. The black dots with red squares or black circles with blue squares are for the incorrectly predicted samples in pre- and post-treatment. 24 Out of the 30 patients with CKD2 of pre-treatment were located in pre-treatment area (80.0% specificity) and 27 out of the 30 patients with CKD2 of post-treatment were correctly grouped (90.0% sensitivity). **g** PCA of two components of five biomarkers. **h** OPLS-DA of five biomarkers. **i** Heatmap of five biomarkers. **j** Diagnostic performances of five biomarkers based on the PLS-DA model. The black dots with red squares and or black circles with blue squares are for the incorrectly predicted samples in pre- and post-treatment. 29 Out of the 30 patients with CKD2 of pre-treatment were located in pre-treatment area (96.7% specificity) and 29 out of the 30 patients with CKD2 of post-treatment were correctly grouped (96.7% sensitivity). These results demonstrated that the five biomarkers show high prediction class probabilities. **k** Analysis of PLS-DA based ROC curves of proteinuria and five metabolites in patients with CKD2 pre- and post-treatment by enalapril. The associated AUC, 95% confidence interval (CI), sensitivity and specificity values were indicated. Student's *t* test was used for the significance of difference between two groups

Significantly increased proteinuria was observed after the treatment with prednisone. Except for tiglylcarnitine, the other four potential biomarkers were significantly restored after patients received a prednisone treatment (Supplementary Fig. 3a, b). PCA, OPLS-DA, and heatmap showed that post-treatment patients with CKD2 could be separated from pre-treatment patients with CKD2 by the combination of proteinuria and five biomarkers (Supplementary Fig. 3c–e). Sensitivity and specificity of the combination of proteinuria and five biomarkers are 80.0 and 83.3%, respectively (Supplementary Fig. 3f). Similar results are also observed by only five biomarkers (Supplementary Fig. 3g–j). ROC analyses show 5-MTP, CSA and acetylcarnitine have a high AUC values compared with the proteinuria (Supplementary Fig. 3k).

Additionally, significantly increased proteinuria was improved after the treatment with Wulingsan. Except for tiglylcarnitine, the other four potential biomarkers were significantly restored after patients received a Oryeongsan treatment (Fig. 5a, b). PCA, OPLS-DA, and heatmap showed that post-treatment patients with CKD2 could be separated from pre-treatment patients with CKD2 by the combination of proteinuria and five biomarkers (Fig. 5c–e). Sensitivity and specificity of the combination of proteinuria and five biomarkers are 93.3 and 86.6%, respectively (Fig. 5f). Similar results are also observed by only five biomarkers (Fig. 5g–j). Interestingly, ROC analyses show proteinuria has a higher AUC values than other five biomarkers by pre- and post-treatment of Wulingsan (Fig. 5k). This results might indicate that Wulingsan show a more beneficial effect on proteinuria improvement in patients with early stage CKD than enalapril or prednisone. These results demonstrated that five metabolites could be considered as biomarkers for diagnosis of early CKD.

**The renoprotective effects of 5-MTP.** Among the five metabolites, 5-MTP was one of the metabolites identified in progressive CKD and strongly associated with eGFR which was consistent with previous finding[10]. Furthermore, the protective effects of 5-MTP were demonstrated in vascular injury[25,26], which were strongly associated with CKD. The anti-inflammatory effects of 5-MTP have been documented by several studies[27], but have not been demonstrated in animals with kidney injury or renal tubular epithelial cells. Thus, we examined the renal protective effects of 5-MTP in vitro and in vivo. To confirm the tissues specificity of 5-MTP, we measured the 5-MTP level in multiple organs. The results demonstrated that 5-MTP was more significantly decrease in the kidney than in other organs such as heart, liver, lung, spleen, prostate and brain in unilateral ureteral obstruction (UUO) mice (Fig. 6a). Next, UUO mice were employed to demonstrate the anti-inflammatory and anti-fibrotic effects of 5-MTP. 5-MTP was administrated at doses of 10 and 100 mg/kg per day by intraperitoneal injection for one week. Hematoxylin and eosin (H&E) and Masson staining indicated that renal inflammation and fibrosis were attenuated by 5-MTP in UUO mice (Fig. 6b). This was associated with increased Nrf2, heme oxygenase-1 (HO-1), NAD(P)H quinone dehydrogenase 1 (NQO-1), inhibitor of kappa B alpha (IκBα) and decreased phosphorylated-IκBα (p-IκBα), nuclear factor kappa B p65 (NF-κB p65), monocyte chemoattractant protein-1 (MCP-1) and cyclooxygenase-2 (COX-2) pointing to attenuation of inflammation in UUO by 5-MTP (Fig. 6c, d). The numbers of CD3[+] and CD68[+] cells in the renal interstitium were significantly decreased by 5-MTP (Fig. 6e, f). Similarly, immunohistochemical examination revealed that expressions of fibronectin and vimentin were attenuated in renal interstitium by 5-MTP (Fig. 6g). The pro-fibrotic proteins collagen I, fibronectin, alpha-smooth muscle actin (α-SMA) were downregulated after treatment with 5-MTP (Fig. 6h, i).

Next, we evaluated the potential inhibitory effects of 5-MTP on inflammation and fibrosis in transforming growth factor beta 1 (TGF-β1)-induced HK-2 cells and lipopolysaccharide (LPS)-induced HMC. Treatment with 5-MTP significantly inhibited expression of inflammatory markers (IκBα, p-IκBα, NF-κB p65, MCP-1 and COX-2) and restored expression of cytoprotective pathway (Nrf2, HO-1 and NQO-1) in both HK-2 and HMC cells, and high concentration 5-MTP showed a more potent anti-inflammatory effects (Fig. 7a–f). Western blot analysis and immunofluorescence staining showed that 5-MTP attenuated the pro-inflammation mediators including NF-κB p65, MCP-1 and COX-2 expression, and increased expression of the cytoprotective anti-inflammatory transcription factor Nrf2 and its target gene products HO-1 and NQO-1 (Fig. 7a–f). 5-MTP also reduced the phosphorylation of IκBα. Treatment with 5-MTP also significantly inhibited the upregulation of the pro-fibrotic proteins collagen I, fibronectin, vimentin, α-SMA (Fig. 7g, h). Immunofluorescence staining showed that 5-MTP attenuated vimentin expression in TGF-β1-induced HK-2 cells (Fig. 7i). Moreover, 5-MTP mitigated epithelial and mesangial cell injury by enhancing expression of epithelial and mesangial cell markers, E-cadherin and Thy1. Taken together these experiments demonstrated that 5-MTP exerts strong anti-inflammatory and anti-fibrotic effects in vitro and in vivo.

**The protective effects of TPH-1 on renal disease.** We hypothesized that upregulation of endogenous 5-MTP by TPH-1 exerts favorable therapeutic effects on renal diseases. TPH-1 is a key enzyme in the synthesis of 5-MTP from L-tryptophan, and knock-down of TPH-1 has been linked to the downregulation of 5-MTP[28]. TPH-1 expression was reduced after kidney injury caused by TGF-β1 and LPS stimulation (Fig. 8a). Knock down of TPH-1 suppressed expression of epithelial cell markers and increased expression of pro-fibrotic markers (Fig. 8b–d). We further investigated the expression of TPH-1 in UUO and renal ischemia/reperfusion injury (IRI) mice and found that TPH-1 expression was significantly downregulated in both UUO and IRI mice (Fig. 8e and Supplementary Fig. 4). TPH-1 overexpression resulted in significant upregulation of 5-MTP in mice (Fig. 8f, g). TPH-1 overexpression decreased of expression pro-inflammatory and pro-fibrotic markers in UUO mice (Fig. 8h–k). In contrast, TPH-1 deficiency exacerbated inflammation and fibrosis in UUO mice (Fig. 8l–o). Taken together, these results identified TPH-1 as a potential therapeutic target for treatment of CKD.

**Discussion**

Through analysis of high-throughput metabolomics data and multi-step validations, we identified 5-MTP, CSA, acetylcarnitine, tiglylcarnitine and taurine which can serve as biomarkers for early stage CKD in addition to proteinuria. The analyses were conducted primarily using advanced bioinformatics approach that combines machine learning techniques and statistic feature reduction method. While the study identified five promising metabolites, other potential metabolites have also been identified. A feature may show very strong effect on the outcome in the univariate analysis but not selected by the LASSO, and vice versa. Univariate analysis is the simplest form of statistical analysis which involves only one variable. The significance of the relationship between any given variable with the outcome is based solely on that variable and does not consider the presence of other variables. While in reality, it is always important not only to evaluate the effect of that variable alone, but also in combination with other variables, some of which may be correlated. In LASSO, when a number of highly correlated variables are associated with the outcome, LASSO tends to select those that are more explanatory among the correlated group. With the presence of the five

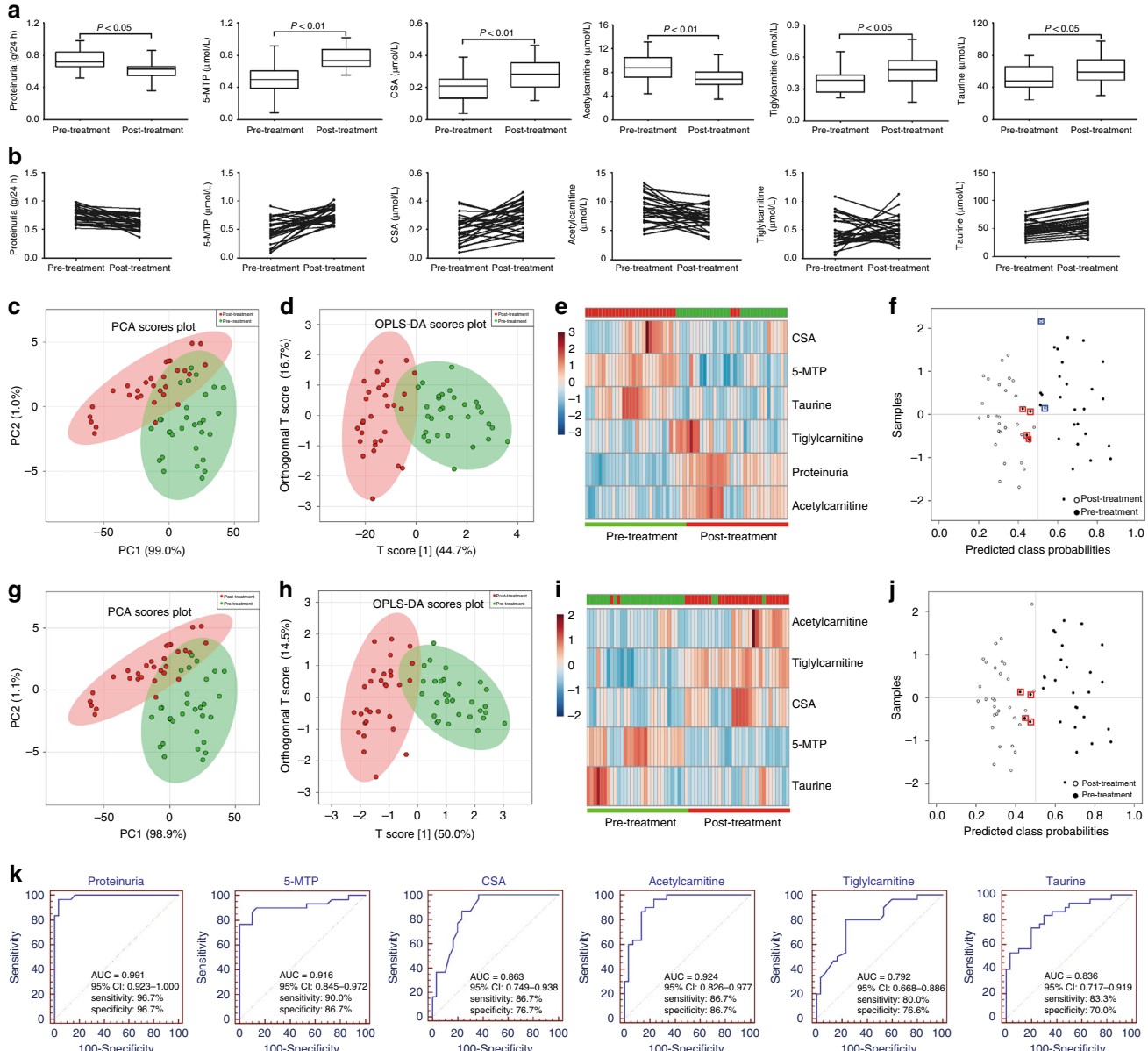

**Fig. 5** Five potential biomarkers were further validated in pre- and post-treatment by Wulingsan. **a** Box plots of levels of five potential biomarkers including 5-MTP, CSA, acetylcarnitine, tiglycarnitine and taurine in patients with CKD2 pre- and post-treatment by Wulingsan. Mean values are presented by horizontal bars. The whiskers indicate the maximum and minimum points. **b** Dots and lines showing changes of five potential biomarkers. **c** PCA of two components of proteinuria and five biomarkers. **d** OPLS-DA of two components of proteinuria and five biomarkers. **e** Heatmap of two components of proteinuria and five biomarkers. **f** Diagnostic performances of two components of proteinuria and five biomarkers based on the PLS-DA model. The black dots with red squares or black circles with blue squares are for the incorrectly predicted samples in pre- and post-treatment. 26 Out of the 30 patients with CKD2 of pre-treatment were located in pre-treatment area (86.6% specificity) and 28 out of the 30 patients with CKD2 of post-treatment were correctly grouped (93.3% sensitivity). **g** PCA of two components of five biomarkers. **h** OPLS-DA of five biomarkers. **i** Heatmap of five biomarkers. **j** Diagnostic performances of five biomarkers based on the PLS-DA model. The black dots with red squares and or black circles with blue squares are for the incorrectly predicted samples in pre- and post-treatment. 26 Out of the 30 patients with CKD2 of pre-treatment were located in pre-treatment area (86.6% specificity) and all the 30 patients with CKD2 of post-treatment were correctly grouped (100.0% sensitivity). These results demonstrated that the five biomarkers show high prediction class probabilities. **k** Analysis of PLS-DA based ROC curves of proteinuria and five potential biomarkers. The associated AUC, 95% confidence interval (CI), sensitivity and specificity values were indicated. Student's $t$ test was used for the significance of difference between two groups

selected metabolites, other metabolites [DG(38:1), DG(42:3), etc.] significant by univariable analysis were not needed to make a robust final model. However, they could also be important markers of altered kidney function, and their role deserves further exploration in future studies.

Our study is first to identify 5-MTP as the most promising biomarker metabolite for detection of early stage CKD. 5-MTP,

an endogenous tryptophan metabolite, decreased in serum but increased in urine in CKD. Few studies reported the changes of 5-MTP in CKD. However, notably, tryptophan show the same trend with 5-MTP in CKD[10,29,30], which may result from reduced reabsorption of tryptophan and 5-MTP in the kidney. 5-MTP is converted by TPH-1[31] and possess anti-inflammatory activity[32]. Furthermore, by suppressing activation of NF-κB and consequent

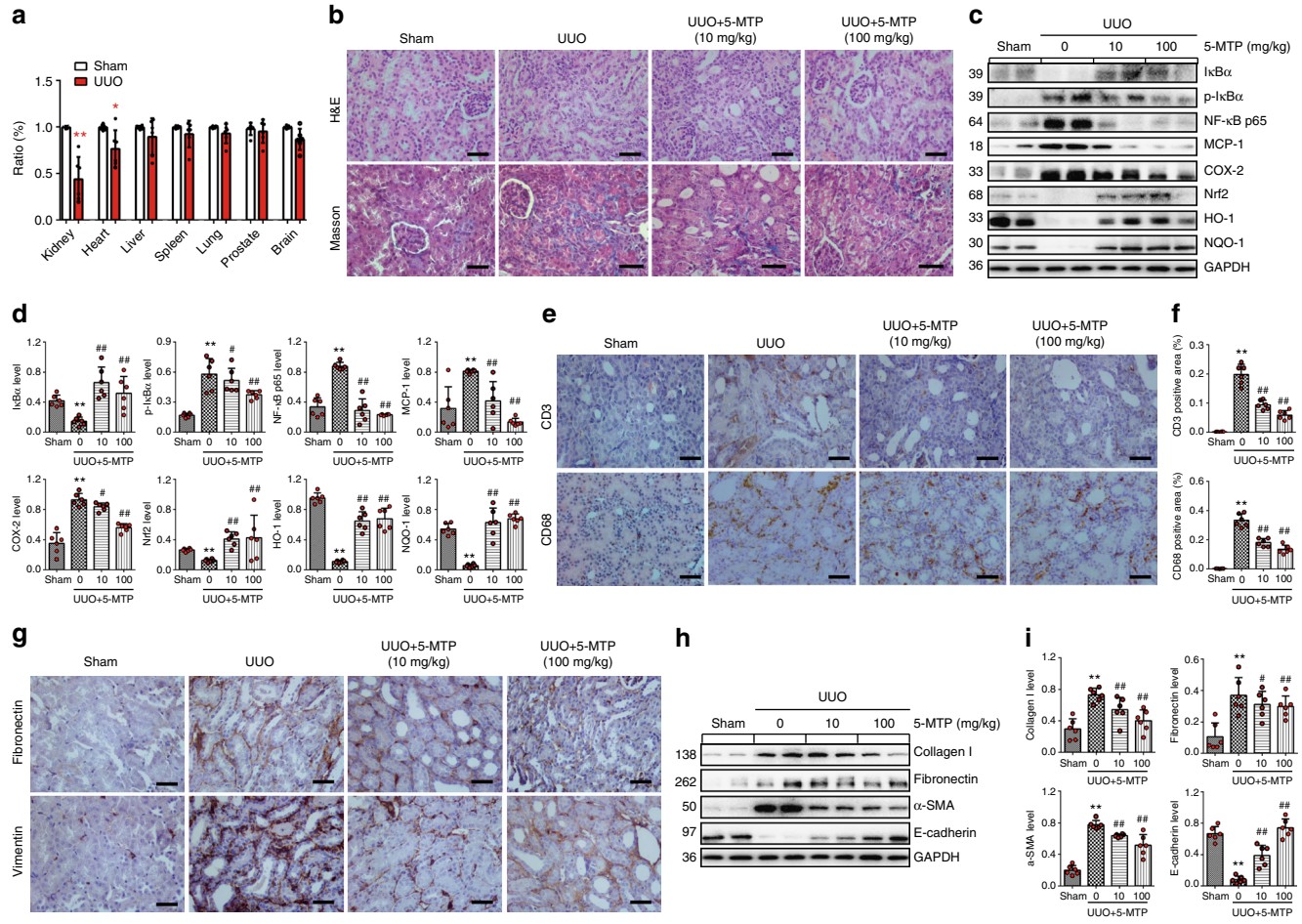

**Fig. 6** The anti-inflammatory and anti-fibrotic effects of 5-MTP in vivo. **a** The level of 5-MTP in the different tissues in UUO mice. *$P < 0.05$, **$P < 0.01$ compared with sham group ($n = 6$). **b** H&E and Masson trichrome staining of kidney tissues of UUO mice. Representative micrographs showed kidney inflammation and fibrotic lesions in indicated groups. Paraffin sections were used for H&E and Masson trichrome staining. Scale bar, 55 μm. **c**, **d** The protein expression and relative quantitative data of IκBα, p-IκBα, NF-κB, and its downstream gene products, COX-2, MCP-1, as well as Nrf2 and its downstream gene products, HO-1 and NQO-1, in indicated groups. *$P < 0.05$, **$P < 0.01$ compared with sham group ($n = 6$). #$P < 0.05$, ##$P < 0.01$ compared with UUO group ($n = 6$). **e**, **f** Immunohistochemical staining of CD3 and CD68 and relative quantitative data in indicated groups. Scale bar, 50 μm. #$P < 0.05$, ##$P < 0.01$ compared with UUO group ($n = 6$). **g** Immunohistochemical staining of fibronectin and vimentin in indicated groups. Scale bar, 50 μm. **h**, **i** The protein expression and relative quantitative data of collagen I, fibronectin, α-SMA and E-cadherin in different groups as indicated. *$P < 0.05$, **$P < 0.01$ compared with sham group ($n = 6$). #$P < 0.05$, ##$P < 0.01$ compared with UUO group ($n = 6$). UUO, unilateral ureteral obstruction. Dot presents the single data results in bar graph. Data are presented as means ± SD. Student's t test was used for the significance of difference between two groups; one-way ANOVA followed by Dunnett's post hoc test for multiple comparisons was used for three or more groups

inhibition of transcriptional activation of COX-2, 5-MTP attenuates production of inflammatory cytokines, thereby alleviates inflammation and tissue injury[33]. In addition, it has been recently suggested that 5-MTP can protect cardiomyocytes against $H_2O_2$-induced oxidative injury, post-myocardial infarction cardiac injury and ventricular remodeling[32,34].

Our investigation of 5-MTP's biological roles using UUO mice and HK-2, HMC cells demonstrated that 5-MTP attenuates the pro-inflammatory factor NF-κB p65, and expression of its target gene products, MCP-1 and COX-2, and increases the anti-inflammatory and antioxidant transcription factor Nrf2 and expression of its target gene products, HO-1 and NQO-1. In addition, treatment with 5-MTP significantly attenuated upregulation of pro-fibrotic proteins collagen I, fibronectin, vimentin and α-SMA and reversed downregulation of E-cadherin and Thy1 in both in vivo and in vitro settings.

TPH-1 is the key regulatory enzyme of 5-MTP[28]. TPH-1 deficiency in the kidney tissue was accompanied by significant upregulation of pro-inflammatory proteins and downregulation

of anti-inflammatory proteins in the cell and animal models. This dysfunction results in the accumulation of extracellular matrix and renal fibrosis. In fact, overexpression of the regulatory enzyme TPH-1 led to upregulation of 5-MTP, and amelioration of kidney injury in our UUO and IRI rats. TPH-1 deficiency exacerbates renal injury through induction of inflammation and fibrosis in UUO mice. In summary, given its critical role in inhibiting renal inflammation and fibrosis, TPH-1 has the potential to be developed as a therapeutic target to treat CKD.

CSA was another important identified metabolite. CSA is synthesized from ureidohomoserine and aspartate by argininosuccinate synthetase[35,36]. Both serum[36] and urinary[37] CSA have been shown to have high sensitivity and specificity as biomarkers in discrimination of hepatocellular carcinoma from liver cirrhosis. In the present study, CSA was identified as one of the major predictive metabolite for early CKD detection. Decreased CSA in patients with early stage CKD was restored by the treatments of enalapril, prednisone and Wulingsan.

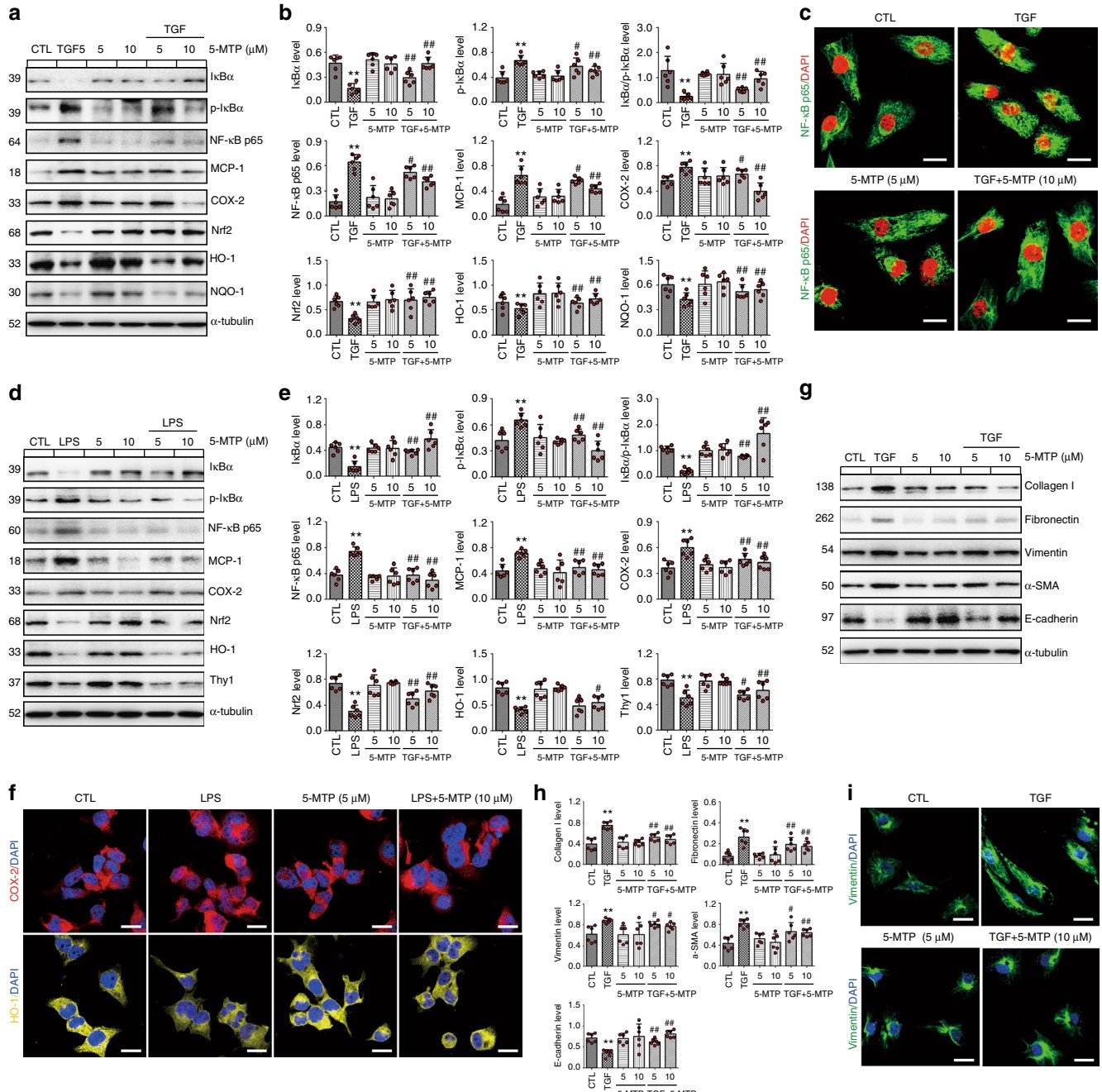

**Fig. 7** The anti-inflammatory and anti-fibrotic effects of 5-MTP in vitro. **a, b** The protein expression and relative quantitative data of IκBα, p-IκBα, NF-κB p65, COX-2 and MCP-1 as well as Nrf2, HO-1 and NQO-1 in HK-2 cells induced by TGF-β1 as indicated. *$P < 0.05$, **$P < 0.01$ compared with CTL group ($n = 6$). #$P < 0.05$, ##$P < 0.01$ compared with TGF-β1-induced group ($n = 6$). **c** Representative immunofluorescent staining of NF-κB p65 in HK-2 cells induced by TGF-β1 as indicated. Scale bar, 25 μm. **d, e** The protein expression and relative quantitative data of IκBα, p-IκBα, NF-κB p65, COX-2 and MCP-1 as well as Nrf2, HO-1 and NQO-1 in HMC induced by LPS as indicated. *$P < 0.05$, **$P < 0.01$ compared with CTL group ($n = 6$). #$P < 0.05$, ##$P < 0.01$ compared with LPS-induced group ($n = 6$). **f** Representative immunofluorescent stainings of COX-2 and HO-1 in HMC induced by LPS as indicated. Scale bar, 25 μm. **g, h** The protein expression and relative quantitative data of collagen I, fibronectin, α-SMA and E-cadherin in HK-2 cells induced by TGF-β1 as indicated. *$P < 0.05$, **$P < 0.01$ compared with CTL group ($n = 6$). #$P < 0.05$, ##$P < 0.01$ compared with TGF-β1-induced group ($n = 6$). **i** Representative immunofluorescent staining of vimentin in HK-2 cells induced by TGF-β1 as indicated. Scale bar, 25 μm. CTL, control; LPS, lipopolysaccharide; TGF, transforming growth factor-β1. Dot presents the single data result in bar graph. Data are presented as means ± SD. One-way ANOVA followed by Dunnett's post hoc test for multiple comparisons was used for three or more groups

Acetylcarnitine could facilitate movement of acetyl-CoA into the matrices of the mitochondria during fatty acid oxidation. Several studies have shown elevation of serum acetylcarnitine in humans and animals with different kidney diseases[38–41]. Accumulation of acetylcarnitine in patients with CKD may lead to the inhibition of carnitine acetyltransferase activity and mitochondrial dysfunction in skeletal muscles by shifting there are action in the opposite direction[42]. Our current study demonstrated that the serum acetylcarnitine level was significantly increased with reduction of renal function in patients with CKD. Elevated serum

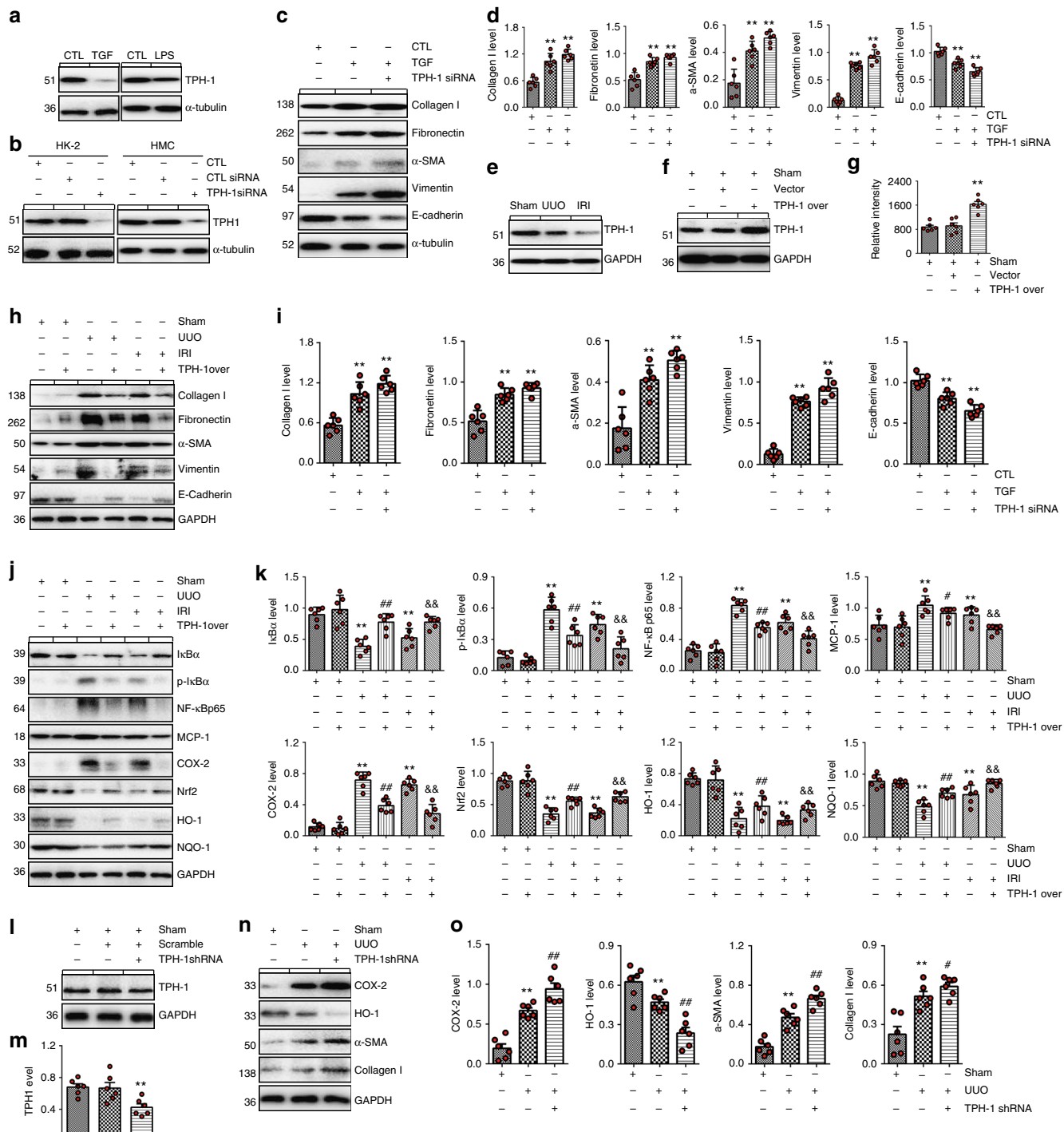

acetylcarnitine levels in patients with CKD may lead to accumulation of acetyl-CoA and depletion of CoA in mitochondria by stimulating the reverse carnitine acetyltransferase reaction. Besides our current study was the first to identify tiglylcarnitine deficiency as a potential biomarker in patients with CKD. However, based on the data collected from Caucasians, Goek et al. and Sekula et al. found that high levels of tiglylcarnitine are linked to low eGFR in patients with CKD[43,44]. It should be noted that the serum samples tested in the Goek et al. and Sekula et al. study were from Caucasians which are different from our samples that were from Han Chinese. The observed difference may be due to differences in the diets and ethnic backgrounds of the study populations. In addition, Liu et al. study was also different from

the present study[45]. In that it studied predominantly diabetic kidney disease (DKD); furthermore, most short to medium chain acylcarnitines differed among the DKD and nonDKD cohorts analyzed in the aforementioned study. We interpret these discrepancies between the cohorts analyzed by Liu et al. and our findings as indicative of a more specific role of tiglylcarnitine in DKD *vs* non-diabetic forms of CKD. Future studies should explore this relationship more fully, since it may lead to a non-invasive marker that differentiates diabetic from non-diabetic forms of CKD in patients.

Taurine is an important endogenous metabolite, which has been shown to be significantly affected in patients with CKD[46,47]. For example, a large population-based study identified changes in

**Fig. 8** The protective roles of TPH-1 in renal injury. **a** The protein expression of TPH-1 in TGF-β1-induced HK-2 cells and LPS-induced HMC, respectively. **b** The expression of TPH-1 in HK-2 and HMC cells after knock-down of TPH-1, respectively. HK-2 cells and HMC were transfected with TPH-1-specific siRNA or CTL siRNA. **c, d** The protein expression of collagen I, fibronectin, α-SMA, vimentin and E-cadherin in HK-2 cells as indicated. *$P < 0.05$, **$P < 0.01$ compared with sham group ($n = 6$). **e** The protein expression of TPH-1 in UUO and IRI mice as indicated ($n = 6$). **f** The protein expression of TPH-1 in mice after injection with lentivirus expressing full-length mouse *Tph1* cDNA (TPH-1 over) or lentivirus containing empty plasmids (vector) ($n = 6$). **g** The relative intensity of 5-MTP in mice as indicated ($n = 6$). *$P < 0.05$, **$P < 0.01$ compared with sham group ($n = 6$). **h, i** The protein expression and relative quantitative data of collagen I, fibronectin, α-SMA, vimentin and E-cadherin in mice as indicated. **j, k** The protein expression and relative quantitative data of IκBα, p-IκBα, NF-κB p65, MCP-1, COX-2, Nrf2, HO-1 and NQO-1 in mice as indicated. *$P < 0.05$, **$P < 0.01$ compared with sham group ($n = 6$). #$P < 0.05$, ##$P < 0.01$ compared with UUO group ($n = 6$). &$P < 0.05$, &&$P < 0.01$ compared with IRI group ($n = 6$). **l, m** The protein expression and quantitative analysis of TPH-1 in mice after injection with lentivirus carrying shRNA against *Tph1* and lentivirus containing nonspecifc shRNA (scramble). **$P < 0.05$ compared with sham mice ($n = 6$). **n, o** The protein expression and quantitative analysis of COX-2, HO-1, α-SMA and collagen I in mice as indicated. *$P < 0.05$, **$P < 0.01$ compared with sham group ($n = 6$). #$P < 0.05$, ##$P < 0.01$ compared with UUO group ($n = 6$). Dot presents the single data result in bar graph. Data are presented as means ± SD. One-way ANOVA followed by Dunnett's post hoc test for multiple comparisons was used for three or more groups. CTL control; IRI ischemia/reperfusion injury, LPS lipopolysaccharide, TGF transforming growth factor-β1, TPH-1 tryptophan hydroxylase-1, UUO unilateral ureteral obstruction

taurine level in patients with early stage CKD[44,48]. Another small cross-sectional study identified taurine deficiency as an important biomarker distinguishing healthy controls form patients with CKD1 to CKD4[49]. Recently, two metabolomics studies have demonstrated a significant reduction of parenchymal taurine in the kidney allograft-form-individuals with low eGFR[50]. The mechanism of CKD-induced taurine deficiency is not clear. Oxidative stress contributes to the pro-inflammatory state in CKD through the NF-κB activation. The improvement in oxidative stress and inflammation in CKD in response to therapy might account for the observed restoration of the taurine level[51].

Our current study has several limitations. We used eGFR to represent kidney function instead of measuring GFR by inulin clearance method which is impractical for a large general population-based study. In addition, the five metabolites were not validated from populations of other races. However, the positive results from cell and animal models suggested possible generalization of 5-MTP in a different population.

In summary, the present study uncovered five metabolites which can separate patients with early stage CKD from healthy control and serve as accurate biomarkers of the early stage kidney disease. These biomarkers can serve as useful supplements to the existing biochemical indicators of kidney disease. And our results have shown that these five metabolites can separate patients with early stage CKD from healthy controls. Mechanistically, the loss of 5-MTP and its regulatory enzyme, TPH-1, contribute to kidney injury by activating NF-κB and inhibiting Nrf2 pathways. Therefore, targeting TPH-1 may serve as a strategy for the treatment of CKD.

## Methods
**Overall participants.** The study was conducted using 2155 patients in four phases. The subjects contain 703 training cohort, 90 validation cohort, 1,248 longitudinal cohort, and 114 drug treatment cohort. The study design is illustrated in Supplementary Fig. 1. Blood and samples were collected between February 2011 and November 2016 from Shaanxi Traditional Chinese Medicine Hospital, Xi'an No. 4 Hospital and Baoji Central Hospital. Patients with acute kidney injury, liver disease, active vasculitis, gastrointestinal pathology or cancer were excluded from the study. All study participants were ethnically Han Chinese. Patients were divided into CKD stage 1, 2, 3, 4 and 5 using creatinine-based eGFR equation[52]. We used the modified CKD-EPI equation to estimate the GFR (eGFR)[53].

**Training cohort.** The initial training dataset contained 703 subjects including 587 CKD patients with five stages (CKD1 = 120, CKD2 = 104, CKD3 = 110, CKD4 = 119, CKD5 = 134) and 116 age-matched normal healthy controls. Healthy controls with no history of kidney disease were enrolled at the same facilities. Healthy controls were excluded if they had any of the following conditions: cardiovascular disease, diabetes, hypertension, kidney dysfunction or use of regular medications. The study was approved by the Ethical Committee and all patients provided informed consent prior to entering the study. Serum samples were obtained after an overnight fasting and serum was separated and stored at −80 °C for biochemical analysis. Blood biochemistry was determined by the clinical laboratory.

**Validation cohort.** An additional 90 serum samples including 30 healthy controls, 30 CKD1 and 30 CKD2 were collected as the external validation cohort. The inclusion-exclusion criteria for external validation cohort are the same as mentioned in the previous paragraph. Patients with acute kidney injury, liver disease, active vasculitis, gastrointestinal pathology or cancer were excluded from the study.

**Longitudinal cohort.** We utilized data from a longitudinal cohort study. A total of 1248 participants were registered in the physical examination center of the four hospitals mentioned above between 2011 and 2016. The participants visited the hospitals once a year. CKD was defined as an eGFR < 60 ml/min per 1.73 m² (CKD stage 3 or worse). Proteinuria was defined as trace or higher levels by dipstick proteinuria in this study. Individuals were excluded from this study if they had CKD at the baseline visit (first examination). Among all 1,248 eligible participants with an eGFR ≥ 60 ml/min per 1.73 m² at baseline, 31 participants developed new-onset chronic renal injuries (eGFR < 60 ml/min/1.73 m²) during the course of the longitudinal study and these individuals were considered as cases[48]. Other 1,217 participants had an eGFR ≥ 60 ml/min/1.73 m² and did not develop chronic renal injuries.

**Drug intervention cohort.** Based on eGFR and 24-hour proteinuria results, an additional 114 patients with CKD2 were studied for further metabolite validation. Forty and thirty-six patients with proteinuria between 0.5 and 1.0 g/24 h were treated by enalapril (10 mg/d) and Wulingsan (0.15 g/kg/d, a renoprotective natural medicine) for six months, respectively. Wulingsan is one of the compund medicines, composed of 5 major ingredients, and has been used for the treatment of various renal diseases.

Out of these patients, 31, 27 and 18 patients had had a kidney biopsy consistent with chronic glomerulonephritis, idiopathic membranous nephropathy and hypertensive nephropathy, respectively. An additional thirty-eight patients had biopsy proven IgA nephropathy. According to the Kidney Disease Improving Global Outcomes (KDIGO) guideline, patients with proteinuria >1.0 g/24 h were treated by decreasing dosages of prednisone 0.8 mg/kg/d in the first two months, 0.6 mg/kg/d during second two months and 0.4 mg/kg/d during the third two months. A total of 228 serum samples were collected from patients with CKD before and after drug treatments.

**Untargeted metabolomics and measurement of metabolites.** The samples were analyzed by a 2.1 × 100 mm ACQUITY 1.8 μm HSS T3 using a Waters Acquity[TM] UPLC system equipped with a Waters Xevo[TM] G2 QTof of MS (Milford, MA, USA). The metabolomic procedure including sample preparation, metabolite separation and detection, data preprocessing and statistical analysis for metabolite identification was performed following previous protocols with minor modifications[10,21,22]. The methods of chromatographic separation and mass spectrometry were described in detail as follow:

The UPLC analysis was performed with a Waters Acquity[TM] Ultra Performance LC system (Waters Corporation, Milford, MA, USA) equipped with a Waters Xevo[TM] G2 QTof MS (Waters MS Technologies, Manchester, UK). Chromatographic separation was carried out at 40 °C on an ACQUITY UPLC HSS T3 column (2.1 × 100 mm, 1.8 μm, UK). The mobile phase consisted of water (A) and acetonitrile (B), each containing 0.1% formic acid. The optimized UPLC elution conditions were: 0–1.0 min, 1.0% B; 1.0–12.0 min, 1.0–99.0% B; 12.0–14.0 min, 99.0–1.0% B and 14.0–15.0 min, 1.0% B. The flow rate was 0.40 ml/min. The autosampler was maintained at 4 °C. Every 1 μl sample solution was injected for each run.

Mass spectrometry was performed on a Xevo[TM] G2 QTof (Waters MS Technologies, Manchester, UK). The scan range was from 50 to 1200 m/z. For positive electrospray mode, the capillary and cone voltage were set at 3 kV and 30 V, respectively. The desolvation gas was set to 600 l/h at a temperature of 450 °C; the cone gas was set to 50 l/h and the source temperature was set to 110 °C. The mass spectrometry was operated in W optics mode with 12,000 resolution using dynamic

range extension. The data acquisition rate was set to 0.1 s, with a 0.014 s interscan delay. Collision energy ramp was 20–30 V. All analyses were acquired using the lockspray to ensure accuracy and reproducibility. Leucine–enkephalin was used as the lockmass at a concentration of 300 ng/ml and flow rate of 5 μl/min. Data were collected in continuum mode, the lockspray frequency was set at 10 s, and data were averaged over 10 scans. All the acquisition and analysis of data were controlled by Waters MassLynx v4.1 software. The mass data acquired were imported to Prognosis QI and Markerlynx XS (Waters Corporation, MA, USA) within the Masslynx software for peak detection and alignment. The resultant data matrices were introduced to the EZinfo 2.0 software (Waters Corporation, Milford, MA, USA).

**UPLC-HDMS method assessment.** The precision and repeatability of the current experiment were tested for assessment of UPLC-HDMS method. Commercial software Prognosis QI and MassLynx developed by Nonlinear Dynamics (Waters Corporation) was used to process the raw data. Missing values were considered undetectable and set to zeros. The repeatability and precision were tested by six reduplicate analyses from the quality control samples and six samples, respectively. Relative standard deviation (RSD%) of retention time and peak area were below 0.52 and 2.9%, respectively. This method showed the good repeatability and precision in this study. Data were log2 transformed and normalized to total signal per patient.

For analytical method assessment, 50 μl each of all the samples were pooled to get a pooled quality control sample that would be tested during the analysis. According to different metabolite polarities and m/z values, ten ions including m/z 161.9852, 391.2812, 381.2846, 764.5300, 441.1958, 820.8169, 208.1360, 133.0861, 486.2534 and 429.2741 were extracted for the assessment for method validation. Injection precision was performed by the continuous analyses of six replicates of the same quality control samples. Injection precision of relative standard deviation % (RSD%) of retention times and peak areas were calculated from tested quality control samples. The six parallel samples were analyzed by UPLC-HDMS to evaluate the sample preparation repeatability. The six parallel samples were performed on the repeatability of sample preparation. Method repeatability RSD% of retention times and peak areas of ten ions were calculated from tested quality control samples. The above-mentioned procedure was performed every day.

**Targeted metabolomics and measurement of serum biomarkers.** Targeted metabolomics was performed on a Waters Acquity UPLC™ BEH C18column (2.1 × 50 mm, 1.7 μm) using a Waters Acquity™ UPLC system. The mobile phase and UPLC gradient elution program of untargeted metabolomics was used for determination of potential biomarkers. The flow rate was 0.3 ml/min. The autosampler temperature and column compartment were set at 4 and 40 °C, respectively. The injection volume was 10 μL.

After the chromatographic separation, tandem mass spectrometry (MS/MS) analysis and data acquisition were carried out on a Waters Micromass® Quattro micro™ mass-spectrometer equipped with an ion interface. The five potential biomarkers including 5-MTP, CSA, acetylcarnitine, tiglylcarnitine and taurine were identified in positive ion mode using untargeted metabolomics, so targeted data detection was performed in the multiple-reaction-monitoring (MRM) in positive ion mode. The reference standards of 5-MTP, CSA, acetylcarnitine, carnitine and taurine were used for metabolite identification (based on the MS/MS and retention time) and quantification analysis. Quantification analysis of five potential biomarkers was performed with MRM of the transitions with m/z 235.1 → 218.1 for 5-MTP, m/z 293.1 → 248.3 for CSA, m/z 204.0 → 145.2 for acetylcarnitine, m/z 244.1 → 145.2 for tiglylcarnitine, and m/z 126.0 → 80.9 for taurine with a scan time of 0.2 s per transition. The MS parameters were optimized. The capillary and cone voltages were 1.0 kV and 30 V, respectively. Source and desolvation temperatures were 110 and 450 °C, respectively. Nitrogen was used as the cone and desolvation gas with a flow rate of 30 and 600 L/h, respectively. The collision energy for five potential biomarkers was 25, 20, 25, 25 and 20, respectively. Data acquisition and processing were performed using MassLynx 4.1 software (Milford, USA).

**Model construction for biomarker discovery.** Data for positive and negative ion modes were analyzed separately, and the result was combined for validation. Our two rounds of LASSO-based variable selection were performed using the R package glmnetcr[54]. The lambda values were chosen based on the Bayesian Information Criterion (BIC). In each round of feature selection, we fitted the penalized ordinal response logistic regression with 100 candidate lambdas and calculated the BIC for each lambda. The optimal model was determined by the lambda corresponding to the minimal BIC. All these procedures are implemented in the package glmnetcr. We performed seven-fold cross validation to obtain an unbiased estimate for the model performance. We randomly divided the dataset into 10 partitions. In each fold of cross-validation we took one part as a test set and combined the rest of nine parts as a training set. We repeated our two-step LASSO-based variable selection with the training set and obtained a prediction model. Then we used this model to prediction on the test set and calculate prediction accuracy. The final accuracy is an average of all the accuracies obtained from all the folds of the cross validation, which is 97.2%.

Ordinal regression (R package "ordinal") and McFadden's pseudo $R^2$ were used to assess the overall model fit using all variables selected by the LASSO method[55]. Heatmap and unsupervised cluster analysis were conducted using R package "heatmap3"[56]. PCA was performed in R. Random forest classification analysis was conducted using the R package "random Forest"[57]. Support vector machine analysis was performed using the R package "e1071".

**Metabolite identification.** The protocol of metabolite identification was carried out based on the reported procedure[58,59]. The significant variables by LASSO regression were identified and annotated by using exact molecular weights, m/z element composition (MassLynx i-FIT software, Waters Corporation, Milford, MA, USA), MS, $MS^E$ fragment, literature comparisons and database searches including Human Metabolome Database (http://www.hmdb.ca), KEGG (http://www.kegg.com) METLIN (https://metlin.scripps.edu), MassBank (https://massbank.eu/MassBank/) and Chemspider (http://www.chemspider.com/). Some metabolite was confirmed by comparison with available reference standards under the same UPLC-HDMS condition (see Supplementary Data 1 for details on metabolite annotation/identification).

To demonstrate the metabolite identification process, we use a potential metabolite with m/z as an example to illustrate the metabolite identification on procedure. First, an accurate mass of the ion (m/z 212.0011) was obtained by high-definition mass spectrometry in negative ion mode. We analyzed the possible adduct types, such as [M−H]⁻, [M+FA-H]⁻, [M+Hac-H]⁻, etc. We preliminarily speculate that the quasi-molecular ion m/z 212.0011 is [M−H]⁻. Second, the assistant software packed in MassLynx i-FIT software (Waters Corporation, Milford, MA, USA) was used to determine the element composition for the peak at m/z 212.0011. MassLynx i-FIT algorithm is used to screen suggested elemental compositions by the likelihood that the isotopic pattern of the elemental composition matches a cluster of peaks in the spectrum, increasing confidence in identified metabolites and simplifying results. Chemical elements and atom amounts including carbon (0–50), hydrogen (amount: 0–500), nitrogen (0–50), sulphur (0–8), helium (0–15), bromine (0–15) and phosphorous (0–10) were used to match m/z data. The lower the i-FIT value, the better the fit. When element compositions were calculated, three possible element compositions of $C_8H_6NO_4S$, $C_5H_{10}NO_4S_2$ and $C_3H_6N_3O_6S$ were obtained. Third, the element composition was compared to those registered in the databases, and only one possible element compositions was determined as $C_8H_6NO_4S$. Fourth, to further identify potential metabolites, the fragmentation pattern from $MS^E$ data collection technique has to be used. With different collision energies, the corresponding $MS^E$ information (similar to MS/MS) was obtained. In negative product ion scan spectrum, the ions like m/z 132.0 and m/z 79.9 were found. We speculated that m/z 79.9 was $SO_3$. Fifth, to define its structure, some databases like Human Metabolome Database (http://www.hmdb.ca), KEGG (http://www.kegg.com) METLIN (https://metlin.scripps.edu), MassBank (https://massbank.eu/MassBank/) and Chemspider (http://www.chemspider.com/) were searched with the molecular mass 212.0011 Da, then several compounds without $SO_3$ group were removed from the candidate list. Finally, m/z 212.0011 was identified as indoxyl sulfate. In addition, indoxyl sulfate was confirmed by comparing this fragmentation pattern with a reference standard. By using the same method described above. A portion of the identified metabolites were confirmed with available reference standards or analogue structure of authentic chemicals by matching their retention time and accurate mass measurement.

**Metabolic pathway analysis.** To map the metabolic pathway of identified 98 metabolites from CKD-associated study, enrichment analysis was performed and visualization of metabolic pathways was obtained by Metscape running on Cytoscape 3.0

**Animal models and treatment strategies.** Male BALB/c mice, weighing 20–22 g, were used to establish UUO model by an established protocol. After general anesthesia, complete UUO was carried out by double-ligating the left ureter by 4–0 silk following the dorsal incision. The ureters of sham operated mice were exposed, but not ligated. Mice were randomly assigned to four groups ($n = 6$): (1) sham control, (2) UUO, (3) UUO + 10 mg/kg 5-MTP, and (4) UUO + 100 mg/kg 5-MTP. 5-MTP at doses of 10 and 100 mg/kg were given to mice by intragastric administration. The four group mice were sacrificed at the 7th day. The ligated kidney were immediately frozen and saved in liquid nitrogen for Western blot and immumohistochemical staining.

Male BALB/c mice, weighing 20–22 g, were used for renal IRI model with 30-min ischemia time. Briefly, under general anesthesia, a dorsal incision was made, and bilateral renal pedicles were clipped for 30 min using bulldog clamp. All procedures followed the ARRIVE Guidelines for Reporting Animal Research.

**Knock-in and knock-down of TPH-1 in vivo.** Lentivirus expressing full-length mouse *Tph1* cDNA (TPH-1 over) and lentivirus containing empty plasmids (vector) were constructed by Sangon (Shanghai, China). Besides, lentivirus carrying shRNA against *Tph*1 and lentivirus containing nonspecifc shRNA (scramble) were also constructed by Sangon. After anesthesia and surgery, mice were laparotomized and injected with recombinant adenovirus vector by a 31 G needle at the lower pole of kidney parallel to the long axis. The 100 μL of saline or lentivirus cocktail ($1 \times 10^5$ IU/μl) were injected into kidney. No toxic effects were founded after treatment with lentiviral vector.

**Cell culture and treatment**. Human kidney proximal epithelial cells (HK-2) and human mesangial cells (HMC) were obtained from the American Type Culture Collection (Manassas, VA, USA). HK-2 and HMC cells were cultured in DMEM/F-12 and DMEM, respectively, supplemented with 10% fetal bovine serum at 37 °C with 5% $CO_2$. HK-2 was treated with 1 μM recombinant human TGF-β1 protein (R&D system, USA), while HMC were treated with 10 μg/ml LPS (L2630, Sigma, USA). The concentrations of 5-MTP (M4001, Sigma, USA) for HK-2 and HMC were 5 and 10 μM for 24 h. After 24-h culture, cells were harvested for next experiments.

The knock-down of TPH-1 were operated by siRNA. TPH-1 siRNA and negative control siRNA, constructed by Sangon (Shanghai, China), were transfected into cells by Lipofectamine RNAiMAX (Invitrogen) according to the manufacturer's guide.

**Immunohistochemical staining**. Paraffin-embedded mice kidney sections (5-μm thickness) were prepared as a routine procedure. The sections were deparaffinized by three xylene washes, hydrated by alcohol and washed with distilled water. The sections were added with three percent hydrogen peroxide to blockade endogenous peroxidase activity. Antigen retrieval was performed by a microwave oven for 15 min in the citrate buffer (10 mM, pH 6.0). Five percent BSA were incubated for 30 min at room temperature. Next, the sections were incubated with the primary antibody at 4 °C overnight. Negative controls were incubated without primary antibodies. After being washed, the sections were incubated with the secondary antibody at room temperature for 2 h. DAB were added to visualize and fifty percent harris hematoxylin were performed for counterstaining, then the sections were mounted by neutral gum. Image analysis was done by using Image-Pro Plus 6.0 software

**Immunofluorescence staining and confocal microscopy**. Cells cultured on coverslips were fixed with 4% paraformaldehyde for 10 min Ten percent goat serum was employed to block nonspecific sites. The slides were incubated with primary antibodies at 4 °C overnight. After three times washed by PBS, the slides were incubated with secondary primary at room temperature for 2 h and 10 min with 4′,6-diamidino-2-phenylindole (DAPI). The slides were mounted with 80% glycerinum in PBS. The slides were stored at −20 °C or examined by a laser-scanning confocal microscope (FV1000, Olympus, Japan) equipped with FV10-ASW 4.0 VIEW (Olympus).

**Western blot analysis**. Protein expression was analyzed by western blot analysis. Protein concentration was measured by Pierce™ BCA Protein Assay Kit (23227, Thermo Scientific, USA). The 20–30 μg of total protein was fractionated by Tris-Glycine resolving gel and transferred to a 0.45 μm polyvinylidene difluoride membrane (10600023, Amersham™ Hybond™, GE Healthcare, USA). After incubated for 1 h in 5% non-fat milk blocking buffer, the membranes were incubated overnight at 4 °C with primary antibody. The secondary antibodies of goat anti-rabbit (1:5000, ab6721, Abcam, USA), goat anti-mouse (1:5000, A21010, Abbkine, USA) or rabbit anti-goat (1:5000, A21110, Abbkine, USA) were incubated with 2 h at room temperature. The membrane was then visualized by enhanced chemiluminescence western blotting detection reagent. Signal intensities from immunoblots were quantified using Image J software (version 1.48v, NIH, Bethesda, MD, USA). Band densities were normalized by α-tubulin or GAPDH expression levels. The uncropped blots were shown in Supplementary Figs 5–13.

The following primary antibodies were employed (dilution): collagen I (1:5000, ab34710, Abcam, USA), α-SMA (1:300, ab7817, Abcam, USA), fibronectin (1:1000, ab2413, Abcam, USA), vimentin (1:1000, ab92547, Abcam, USA), E-cadherin (1:500, ab76055, Abcam, USA), p-IκBα (1:2000, 2859, Cell Signaling Technology, USA), IκBα (1:2000, 4812, Cell Signaling Technology, USA), NF-κB p65 (1:1000, ab16502, Abcam, USA), MCP-1 (1:1000, ab7202, Abcam, USA), COX-2 (1:1000, ab62331, Abcam, USA), Nrf2 (1:1000, ab31163, Abcam, USA), HO-1 (1:1000, ab68477, Abcam, USA), NQO-1 (1:1000, ab28947, Abcam, USA) and TPH-1 (1:500, ab52954, Abcam, USA). Glyceraldehyde 3-phosphate dehydrogenase (GAPDH, 1:5000, 10494-1-AP) and α-tubulin (1:1000, 11224-1-AP) were purchased from Proteintech Company (Wuhan, China).

**Statistics**. The number of replicates was 6 per group for each data set and results were presented as mean ± standard deviation (SD) unless stated otherwise. GraphPad Prism (GraphPad software, San Diego, CA, USA) was used for statistical analysis of the data. The significance of the differences between two groups was analysed using unpaired Student's $t$ test, and multiple comparisons was performed by one-way analysis of variance (ANOVA) followed by Dunnett's post hoc test. All tests were two-tailed. $P < 0.05$ was considered statistically significant.

**Study approval**. The part of patient study was approved by the Ethical Committee and all patients provided informed consent prior to entering the study, and all clinical investigation have been conducted according to the principles expressed in the Declaration of Helsinki. The present study has complied with all relevant ethical regulations. The sample collection was approved Shaanxi Traditional Chinese Medicine Hospital (Permit Number: SXSY-235610). For human subjects, written informed consent was received from participants prior to inclusion in the study.

This part of animal study was carried out in strict accordance with the recommendations in the Guide for the Care and Use of Laboratory Animals of the State

Committee of Science and Technology of the People's Republic of China. The present study has complied with all relevant ethical regulations. All protocols were approved by the Committee on the Ethics of Animal Experiments of the Northwest University (Permit Number: SYXK 2010-004). All surgery was performed under uretane anesthesia, and all efforts were made to minimize suffering. All procedures and care of the rats were in accordance with the institutional guidelines for animal use in research.

**Reporting Summary**. Further information on experimental design is available in the Nature Research Reporting Summary linked to this article.

## Code availability
Majority of the data analyses were performed using R x64 3.4.2. All R codes written for this manuscript are available from the corresponding author upon request.

## Data availability
The data that support the findings of this study are available from the corresponding author upon reasonable request.

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

## Acknowledgements

This study was supported by the National Nature Science Foundation of China (Nos 81872985, 81673578).

## Author contributions

Y.-Y.Z., Y.G. and D.Q.C. conceived and designed the project, and managed the study. W.S., L.Z. and X.-H.C. made clinical diagnosis, recruited subjects and performed intervention. X.Y.Y., J.R.M., S-X.M., J.Z., Y.Z., Y.Q.S. and M.-X.M. collected samples and clinical phenotypes. Y.G., C.P.A., H.Y., D.C.S., S.L.Z., H.K. and F.Y. performed bioinformatics analyses. Y.-Y.Z., D-Q.C., G.C., H.C. and X.-R.L. performed metabolomics profiling and data analysis. D.Q.C., L.C., M.W. and D.L. conducted animal and cell experiments. Y.-Y.Z., Y.G., D.-Q.C. and N.D.V. wrote the manuscript. Y.-Y.Z., Y.G., D.-Q.C., N.D.V., S.G.Z. and G.P.B. contributed to text revision and discussion.

## Additional information

**Competing interests:** The authors declare no competing interests.

