## [Peer Review File · Nature Communications]

Reviewers' comments:

Reviewer #1 - expert in CKD (Remarks to the Author):

This is a comprehensive study linking serum metabolites with CKD stage through discovery and validation studies (n=2155 patients in total in 4 different cohorts) coupled with in vitro and in vivo studies that support a renoprotective role for one metabolite - 5-methoxytryptophan or 5-MTP – in the pathogenesis of progressive kidney injury. They report that the enzyme tryptophan hydroxylase-1 (THP-1) that regulates 5-MTP synthesis is downregulated in kidney cell cultures and mouse models of chronic kidney injury. A significant correlation between plasma 5-MTP levels and eGFR was also reported by Chen et al (Redox Biol 12:505, 2017) and several publications (65 total in a PubMed search for 5-MTP) provide further plausibility for its cellular protective role.

GENERAL COMMENTS

The paper begins with an aim of identifying metabolic biomarkers to improve the diagnosis and management of CKD, especially the early stages (CKD1 and CKD2 versus normal where serum creatinine levels may be less useful). The data in Figure 1 suggest that 5-MTP may prove useful. There is one potentially confounding variable in all 4 cohorts – they all are characterized as having significant proteinuria. Whether these findings would apply to non-proteinuric CKD cohort is unclear. Furthermore, the results reported in Figure 4 comparing pre- and post- enalapril treatment (duration unclear) show significant reductions in proteinuria, significant increases in serum 5-MTP level but changes in eGFR are not reported, leaving one to wonder if no differences were observed.

The potential strength of the study is the additional studies designed to investigate biological plausibility of a renal protection for 5-MTP. There are numerous data presented that are intriguing but the experimental design is unnecessarily complicated, leading the reader to get bogged down in too many details. It would be better to use simple systems and forget the addition of drugs (enalapril, picroic acid). These drugs have complex effects and trying to decipher the relative contributions of its effects on 5-MTP should be deferred while the in vivo studies on the direct effects on 5-MTP need further development.

SPECIFIC COMMENTS

1. What cells in the kidney synthesize 5-MTP and is there a way to localize its production in tissue? Does it co-localize with THP-1? THP-1 levels appear to decline with renal injury yet urinary 5-MTP levels (Figure 1) increase. How is this explained? The in vivo studies in Figure 5 (B-H) evaluating

the therapeutic effects of exogenous 5-MTP (2 different doses) are important. How were the doses selected? What is known about its pharmacokinetics? Are there likely kidney cell-specific effects, noting that the literature reports its effects on fibroblasts that are also key cells in kidney fibrosis? The data on CD3 and CD68 cells should be quantitated. The mesangial cells in vitro studies could be deferred to better align with the UUO model that is not known to be a model of primary glomerular injury.

2. Figure 5K (HK-2 cells). There are inconsistencies in the methods. Were these cells only stimulated with LPS and not TGF-beta?

3. A brief mention should be made in the methods section about how the metabolomics studies controlled for differences in loading/serum volume etc. What is the degree of intra-individual variability in the levels of the 5 metabolites of interest?

4. Figure legends should specify n=?. Was it really 6 for every single experiment? Many of the bar graphs lack an explanation of the groups they represent; presumably they align with the Western blots but each figure should be understandable without needed to refer to another one.

5. Figure 6 would be easier to follow if the in vitro and in vivo studies are separated. Eliminating the ENA and PAA studies altogether is recommended. The in vivo studies using TPH-1 knock-in (6F-H) and knockdown (6S-T) need to be expanded with important quality control studies, the latter ideally without the presence of the drugs. How do we know that the protocol used achieved delivery to the appropriate kidney targets, altered THP-1 activity and 5-MTP synthesis?

6. The discussion can minimize that paragraphs on the other 4 metabolites that were not investigated further in this study.

Reviewer #2 - expert in metabolomics (Remarks to the Author):

This manuscript presents a translational metabolomic study of chronic kidney disease, combining profiling of several clinical studies (training cohort, n=703, replication, n=90,,longitudinal follow up, n=1,248, drug treatment n=114) followed by in vivo experiments in the mouse.

Using ultra-performance liquid chromatography coupled to mass spectrometry (UPLC-MS), the investigators identify 5 metabolites associated with CKD, which they further study in subsequent clinical and pre-clinical validations (mouse models, kidney cell culture with knock-ins and knock-downs). In particular, they focus on 5-methoxytryptophan (5-MT), as one of their most promising candidates.

In short, 5-MT decreases with CKD severity and the investigators show that 5-MT reduces inflammation and fibrosis, they then target the biosynthetic enzyme for 5-MT, TPH-1 and successfully demonstrate that variation in inflammation and fibrosis.

Whilst this is an well-rounded report with multiple validations, I have the following concerns which prevent publication of the paper in its current form

Major comments:

- candidate identification and prioritization: The identification of the 5 candidate metabolites in the "final model" presented in Table 2 is unclear. In particular the rationale for focussing on 5-MT when acetylcarnitine is more significant should be emphasised
- novelty: the anti-inflammatory roles of 5-MT are already documented
- reproducibility: the UPLC-MS methods are not explicit and the statistics section does not mentions whether multiple testing corrections are applied or not. if so, I would expect the investigators to clarify which procedure was used.

Minor comments:

- Table S1-4 are redundant, it seems only 1 table is needed.
- The retention times and m/z of the metabolites should be listed
- Multivariate methods are mentioned but with little or no explanation on the parameters used.
- Figures seem to be organised in a systematic way, unfortunately the files I have to review are not legible, which makes the assessment of barplots almost impossible

Reviewer #3 - expert in metabolomics (Remarks to the Author):

In their manuscript “Metabolites strongly associated with estimated glomerular filtration rate and anti-fibrotic effect of 5-methoxytryptophan” the authors used a non-targeted metabolomics approach in serum from 703 subjects for the identification of potential biomarkers indicating early stages of chronic kidney disease (CKD). Five metabolites, including 5-methoxytryptophan (5-MTP) were highlighted and validated as biomarkers in independent cohorts and study designs (cross-sectional, longitudinal, pre-/post treatment). In addition, effects of modulating 5-MTP by supplementation or manipulation of TPH-1 expression were further evaluated using an animal model. Based on their analyses, the authors concluded that (i) the highlighted five metabolites are good biomarkers for early kidney disease (stages 1 and 2), (ii) 5-MTP is renoprotective, and (iii) TPH-1 might be a promising drug target for treating CKD.

While several studies identified metabolites that allow early detection of kidney function decline by analyzing metabolomics data in large cohorts previously, I appreciate the depth of validation and experimental follow-up of the metabolites highlighted in the present work, which I would consider the study's major strength. Unfortunately, the approaches used for identifying the five highlighted metabolites are unclear and not described in sufficient detail and clarity in my opinion. Based on the provided information, I actually found it very difficult to judge the validity of the approach. Moreover, I would have expected a more thorough discussion of the results in the light of the related previous studies on the association of eGFR and metabolites in population-based cohorts, at least in the case of contradicting results, such as the opposite effect directions observed for tiglylcarnitine.

Please see below for more detailed comments:

1) Description of the stepwise feature selection using LASSO is unclear and needs much more details to allow judging the validity of the approach and its results:

a. The description of model construction is confusing and extremely spread in the supplement:

- The 8 lines of description provided in the Supplementary Material on top of pg. 1 under the title “Model construction for variable selection and biomarker discovery”, gives the impression that all 25,109 features went into the construction of the LASSO model (“In our study, we included 703 subjects in our initial dataset and 25109 features. By using LASSO regularization method, we obtained 98 metabolites from 25109 features.”). The very same text is repeated on pg. 4 of the supplement (“Model construction for variable selection and biomarker discovery”)!

- From Figure S1, and the paragraph entitled “UPLC-MS analysis” (?), I understand that you built separate models for positive mode features, negative mode features and clinical features (which ones?) in the first round of feature selection. The second round of feature selection is basically not described anywhere beyond Figure S1.

- On pg 4., in the sections entitled “Model development, data analysis and differential metabolite identification”, there is one sentence informing the reader that LASSO was performed using the glmnetcr package and 10-fold cross validation. Unfortunately, results from the cross-validation are not provided anywhere.

- On pg. 5 of the supplement in the section “Biomarker selection and validation” (?) a short explanation for LASSO is provided stating how the LASSO method works in principle: “LASSO is a feature selection technique that uses least absolute shrinkage, ...”.

b. Some details that are important to allow following or repeating your approach are missing completely, for example: How did you select lambda in your two rounds of LASSO-based feature selection? How does the choice of lambda affect model fit? Which 17 clinical features were used?

c. The results from the multivariate ordinal regressions in Tables S1, S2, and S3 are not very informative. In my opinion, information on the outcome of the LASSO models, in particular the final one, from which the five metabolites were taken, would be much more relevant. In addition, the effect sizes from univariate tests could be provided for each metabolite, which are especially interesting in the cases of highly correlated metabolites, for which the effect seems to be kind of “spread” over several compounds in the multivariate model. Some further questions/comments regarding Tables S1, S2, S3:

- Did you use any relevant covariates such as age, sex, BMI, and blood pressure, diabetes etc. in the models?

- Listing effects in a format so that most of them appear as 0.00 like in Table S3 is not very informative.

- Looking at Table S3, it is a bit surprising that none of the lipids (e.g. PC(33:3), DG(42:3), TG(68:7)) that showed low p-values in the ordinal regression was selected in the final LASSO. Is there any good explanation for that?

d. My understanding is that metabolites measured in urine were not used for building the models. I would be curious to know why? In principle urine is more easily accessible than serum, which could be an advantage for actual clinical use.

2) Information provided for the metabolomics approaches and the pre-processing of the data is extremely sparse. Though I agree that readers can be referred to previous publications for details about the methods, the analytical procedures and main steps in data processing, which are important for the interpretation of results, should at least be delineated in the Supplementary

Methods in my opinion. Moreover, all the mentioned “minor modifications” (Suppl. Mat., pg. 3) from the published protocol should be clearly described. In addition:

a. Please provide information on the level of confidence in your metabolite identification according to the guidelines of the Metabolomics Standards Initiative (MSI).

b. From Figure S2B it looks like missing values in the metabolomics data were set to zero or imputed with a low value. Maybe I have missed it, but I could not find any comment on the handling of missing values in the manuscript or the supplement.

3) Discussion:

a. While, in the present study, low levels of tiglycarnitine (C5:1) were found to associate with higher stages of CKD, various previous studies reported high levels of C5:1 to be linked to low eGFR [Goek et al., *Am J Kidney Dis.*, 2012; PMID: 22464876. Sekula et al., *J Am Soc Nephrol.*, 2016; PMID: 26449609], and diabetic kidney disease [Liu et al., *Kidney Int Rep.*, 2017; PMID:29142974.]. Please discuss!

b. Pg 7, second paragraph: acylcarnitine (i.e., the sum of molecules with a fatty acid (various lengths and desaturation) linked to carnitine) and acetylcarnitine (C2) appear to be used like synonyms in this paragraph. It was not always clear to me to which of the two you actually wanted to refer.

4) Validation models:

a. When analyzing the data from the drug treatment cohort, wouldn't it be more interesting to examine how changes in the five metabolites correlate with the improvement of proteinuria instead of “predicting” pre-/post treatment?

b. Pg 4: Figure 3K does not exist. Did you mean 3G?

5) Co-abundance/correlation analysis:

a. Metabolites are typically highly correlated even if they are only indirectly connected. Therefore, co-abundance networks based on metabolomics data usually do not reflect metabolic pathways well.

b. Figure S3A/Table S4: In the Figure caption (and table header), you wrote that you analyzed correlation between the five “potential biomarkers and 337 metabolites”. However, my understanding was that you only identified metabolites represented by the selected features (98). Do you mean 337 metabolite pairs with correlation >0.9 (assuming that “P value” in the header of Table S3 is a mistake)? The caption also reads as if all correlations > 0.9 are shown in Figure S3A. If so, why is acetylcarnitine only linked to one metabolite in the figure but has much more correlations >0.9 according to Table S4?

c. In general, I cannot see how the analysis of co-abundance/correlation of the measured metabolites or the pathway enrichment analysis adds to or supports the claim of the paper. Results are only mentioned with one sentence in the manuscript (pg. 3: “Pathway and co-abundant analyses shed light on potential metabolic process of CKD (Supplementary Material and Figure S2, S3)”). Maybe I missed it, but I could not find any description or discussion related to these “potential metabolic processes”. I would suggest to either describe them explicitly in the Results and Discussion sections or to remove the analysis completely from the manuscript.

6) Is there any rational behind focusing on TPH-1 but not on other genes in the pathway of 5-MTP such as ASMT2 or IDO2?

7) Typo on pg. 7, first line: THP-1 should be TPH-1.

Responses to reviewers' comments NCOMMS-18-11249-T

We thank the reviewers for their thoughtful and constructive comments, which have guided the revision of this manuscript (Manuscript ID NCOMMS-18-11249-T). Major changes in the manuscript have been highlighted in yellow for easy identification. The concerns have been addressed as follows:

Reviewers' comments:

To Reviewer #1 - expert in CKD (Remarks to the Author):

This is a comprehensive study linking serum metabolites with CKD stage through discovery and validation studies (n=2155 patients in total in 4 different cohorts) coupled with in vitro and in vivo studies that support a renoprotective role for one metabolite - 5-methoxytryptophan or 5-MTP – in the pathogenesis of progressive kidney injury. They report that the enzyme tryptophan hydroxylase-1 (THP-1) that regulates 5-MTP synthesis is downregulated in kidney cell cultures and mouse models of chronic kidney injury. A significant correlation between plasma 5-MTP levels and eGFR was also reported by Chen et al (Redox Biol 12:505, 2017) and several publications (65 total in a PubMed search for 5-MTP) provide further plausibility for its cellular protective role.

GENERAL COMMENTS

The paper begins with an aim of identifying metabolic biomarkers to improve the diagnosis and management of CKD, especially the early stages (CKD1 and CKD2 versus normal where serum creatinine levels may be less useful). The data in Figure 1 suggest that 5-MTP may prove useful. There is one potentially confounding variable in all 4 cohorts – they all are characterized as having significant proteinuria. Whether these findings would apply to non-proteinuric CKD cohort is unclear. Furthermore, the results reported in Figure 4 comparing pre- and post- enalapril treatment (duration unclear) show significant reductions in proteinuria, significant increases in serum 5-MTP level but changes in eGFR are not reported, leaving one to wonder if no differences were observed.

Response: Currently, proteinuria and eGFR decline are mainly biomarkers for the diagnosis and prognosis of renal function. In the clinic, patients with proteinuria and eGFR decline have high morbidity and mortality. Although most of our samples were from patients with CKD and proteinuria, a small number of samples from patients with CKD but without proteinuria were also collected. Patients were treated by enalapril for six months. The eGFR values of pre- and post- enalapril treatment have been provided in the revised manuscript, under the “**Validation of drug treatment**” section.

The potential strength of the study is the additional studies designed to investigate biological plausibility of a renal protection for 5-MTP. There are numerous data presented that are intriguing but the experimental design is unnecessarily complicated, leading the reader to get bogged down in too many details. It would be better to use simple systems and forget the addition of drugs (enalapril, piroxic acid). These drugs have complex effects and trying to decipher the relative contributions of its effects on 5-MTP should be deferred while the in vivo studies on the direct effects on 5-MTP need further development.

Response: Thanks for your suggestion. The reference to additional drugs was removed in the revised manuscript.

SPECIFIC COMMENTS

1. What cells in the kidney synthesize 5-MTP and is there a way to localize its production in tissue? Does it co-localize with TPH-1? TPH-1 levels appear to decline with renal injury yet urinary 5-MTP levels (Figure 1) increase. How is this explained? The in vivo studies in Figure 5 (B-H) evaluating the therapeutic effects of exogenous 5-MTP (2 different doses) are important. How were the doses selected? What is known about its pharmacokinetics? Are there likely kidney cell-specific effects, noting that the literature reports its effects on fibroblasts that are also key cells in kidney fibrosis? The data on CD3 and CD68 cells should be quantitated. The mesangial cells in vitro studies could be deferred to better align with the UUO model that is not known to be a model of primary glomerular injury.

Response: We thank the distinguished reviewer for careful reviewing our manuscript. 5-MTP is a low-molecular-weight metabolite and the molecular weight is 234 Da. No commercial antibodies for 5-MTP is available for localizing 5-MTP in the kidney, so we cannot localize 5-MTP. However, we localized TPH-1 by immunohistochemical analysis. The results demonstrated that in normal condition, TPH-1 was highly expressed in the kidney tissues, but once an injury occurred to kidney, TPH-1 expression was significantly decreased. The immunohistochemistry results have been added in **Figure S5 of Supplementary Material**.

TPH-1 staining of kidney tissues in patients with CKD (A) and mouse models of UUO and IRI (B).

In the current manuscript, our study focuses on the serum rather than urinary metabolomics. When we submitted the manuscript to Nature Medicine, the study of urinary samples was involved in the submitted manuscript. However, we did not describe and discussed the urinary study in the text of submitted manuscript to Nature Communications. We are very sorry that we forgot to remove the urinary study in Figure 1. According to reported studies, 100 mg/kg 5-MTP for mice model and 10 μ M for cultured cells confer good protective effects on several diseases (Proc Natl Acad Sci U S A 2012;109(33):13231-13236; Sci Rep 2016;6:25374). Besides, we also found 10 mg/kg and 100 mg/kg 5-MTP treatment could inhibit inflammation and fibrosis, so the doses of 10 and 100 mg/kg were selected to treat UUO model and the concentration of 5 and 10 μ M were selected to treat cultured cells.

Thanks for your question. In the present study, our aim was focused on clarifying the therapeutic effects of 5-MTP rather than pharmacokinetic study of 5-MTP. In addition, the data on CD3 and CD68 cells have been quantitated and located in **Figure 5F**.

Since many studies have demonstrated that mesangial cells are involved in inflammatory response, LPS-treated mesangial cells were usually used to align with inflammation, rather than align with primary glomerular injury (Kidney Int 2018;93(1):95-109; Nephrol Dial Transplant 2009;24(6):1753-1758; J Am Soc Nephrol 2010;21(1):73-81). Based on metabolomic approach, we identified 5-MTP as an important endogenous metabolite for diagnosis and prognosis in progressive CKD. Further, our aim mainly focused on investigating the pharmacological effects of 5-MTP's renoprotection

based on *in vivo* and *in vitro* study. As you said, the UUO model is a well established model of experimental renal injury characterized by progressive tubulointerstitial fibrosis, and is not a model of primary glomerular injury. Our findings demonstrated that 5-MTP treatment can ameliorate renal interstitial fibrosis, inhibit I κ B/NF- κ B signaling pathway and enhance Keap1/Nrf2 signaling pathway in UUO mice. TGF- β 1-treated HK-2 cells and LPS-treated mesangial cells were used to investigate the anti-fibrotic and anti-inflammatory effects of 5-MTP *in vitro*, respectively. We found that 5-MTP treatment can inhibit I κ B/NF- κ B signaling pathway and enhance Keap1/Nrf2 signaling pathway in both TGF- β 1-induced HK-2 cells and LPS-induced mesangial cells. Taken together, our findings demonstrated that 5-MTP can reduce renal injury by inhibiting inflammation and oxidative stress and attenuating fibrosis.

2. Figure 5K (HK-2 cells). There are inconsistencies in the methods. Were these cells only stimulated with LPS and not TGF-beta?

Response: Sorry for confusion. The label of Figure 5K was not correct. In Figure 5, HK-2 cells were treated with TGF- β 1. We have changed it.

3. A brief mention should be made in the methods section about how the metabolomics studies controlled for differences in loading/serum volume etc. What is the degree of intra-individual variability in the levels of the 5 metabolites of interest?

Response: Ultra-performance liquid chromatography coupled with quadrupole time-of-flight synapt high-definition mass spectrometry (UPLC-QTOF/HDMS) was used to analyze serum samples. UPLC has an auto-sampler. Every 1 μ L sample solution was injected for each run by an auto-sampler. To minimize thermal degradation of the metabolites when waiting to be analyzed, the auto-sampler compartment was set at 4°C throughout the analysis. Analytical method assessment including precision and repeatability has been described in the Supplementary Material, under the “**Analytical method assessment**” section.

4. Figure legends should specify n=? Was it really 6 for every single experiment? Many of the bar graphs lack an explanation of the groups they represent; presumably they align with the Western blots but each figure should be understandable without needed to refer to another one.

Response: Thanks for your kind suggestion. We have added n=6 in each figure legend, and every single experiment was repeated 6 times. The explanations of bar graphs have been added.

5. Figure 6 would be easier to follow if the in vitro and in vivo studies are separated. Eliminating the ENA and PAA studies altogether is recommended. The in vivo studies using TPH-1 knock-in (6F-H) and knockdown (6S-T) need to be expanded with important quality control studies, the latter ideally without the presence of the drugs. How do we know that the protocol used achieved delivery to the appropriate kidney targets, altered THP-1 activity and 5-MTP synthesis?

Response: We thank the distinguished reviewer for the excellent suggestion. The studies associated with ENA and PAA in Figure 6I-T was removed. TPH-1 was significantly increased in mice after injection with lentivirus expressing full-length mouse Tph1 cDNA (TPH-1 over) compared to injection with lentivirus containing empty plasmids (vector) (Figure 7F). The overexpression of TPH-1 resulted in increased level of 5-MTP, which was detected by using mass spectrometry (Figure 7G).

Protein expression of TPH-1 and relative intensity of 5-MTP in indicated mice.

In addition, after injection with lentivirus carrying shRNA against Tph1 (TPH-1 shRNA), the expression of TPH-1 significantly decreased in mice compared to injection with lentivirus containing non-specific shRNA (scramble) (Figure 7L,M). TPH-1 deficiency resulted in the upregulation of pro-inflammatory COX-2, pro-fibrotic α -SMA and collagen I, and the downregulation of anti-inflammatory HO-1 in UUO mice, indicating that TPH-1 deficiency exacerbated renal injury (Figure 7N,O).

(L, M) Protein expression and quantitative analysis of TPH-1 in indicated mice. (N, O) Western blot analyses and quantitative data show renal expression of COX-2, HO-1, α -SMA and collagen I in indicated mice.

6. The discussion can minimize that paragraphs on the other 4 metabolites that were not investigated further in this study.

Response: Thanks for your suggestion. The discussion of the other 4 metabolites has been minimized.

Reviewer #2 - expert in metabolomics (Remarks to the Author):

This manuscript presents a translational metabolomic study of chronic kidney disease, combining profiling of several clinical studies (training cohort, n=703, replication, n=90, longitudinal follow up, n=1,248, drug treatment n=114) followed by in vivo experiments in the mouse.

Using ultra-performance liquid chromatography coupled to mass spectrometry (UPLC-MS), the investigators identify 5 metabolites associated with CKD, which they further study in subsequent clinical and pre-clinical validations (mouse models, kidney cell culture with knock-ins and knock-downs). In particular, they focus on 5-methoxytryptophan (5-MT), as one of their most promising candidates.

In short, 5-MT decreases with CKD severity and the investigators show that 5-MT reduces inflammation and fibrosis, they then target the biosynthetic enzyme for 5-MT, TPH-1 and successfully demonstrate that variation in inflammation and fibrosis.

Whilst this is a well-rounded report with multiple validations, I have the following concerns which prevent publication of the paper in its current form

Major comments:

- Candidate identification and prioritization: The identification of the 5 candidate metabolites in the "final model" presented in Table 2 is unclear. In particular the rationale for focusing on 5-MT when acetylcarnitine is more significant should be emphasized

Response: We thank the distinguished reviewer for careful review of our manuscript. Table 2 presented the final models which contained 5 metabolites. The detail on the identification of these five metabolites was described in the Supplementary Material of the paper, under the “**Metabolite Identification**” section. We first used an ordinal LASSO model to identified 98 metabolites that were closely associated with eGFR. We felt that a 98 metabolites panel is too large to be used clinically. Thus, we conducted a second round of ordinal LASSO analysis using these 98 metabolites, and the model reduced the metabolites to five, which were denoted in Table 2. 5-MTP was selected as the follow up instead the other four metabolites. TPH-1 could convert *L*-tryptophan to 5-MTP, which was a focus of previously studies. Furthermore, the protective effects of 5-MTP has been identified in various diseases such as vascular injury (Sci Rep. 2016 May 5;6:25374; Arch Biochem Biophys. 2014 Feb 1;543:15-22) and endothelial barrier dysfunction (PLoS One. 2016 Mar 22;11(3):e0152166; Circ Res. 2016 Jul 8;119(2):222-36). So far, our current study first identified 5-MTP in progressive chronic kidney disease. Given the previous interests in 5-MTP and our own study with 5-MTP, we decided to follow up on the renoprotective effects of 5-

MTP. We have made the rationale more clear in the revised text (Result -> The Renoprotective Effects of 5-MTP subsection)

- Novelty: the anti-inflammatory roles of 5-MT are already documented

Response: We agree with the reviewer that there are previous documentations of the anti-inflammatory effects of 5-MTP, but rarely in the animal model of renal injury and cell models. So our current study first identified 5-MTP deficiency in progressive chronic kidney disease and further investigated its renoprotective effects. Our *in vitro* and *in vivo* results provided strong evidence of 5-MTP's anti-inflammatory effects in kidney for the first time. Of note, we also reported the anti-fibrotic effects of 5-MTP for the first time.

- reproducibility: the UPLC-MS methods are not explicit and the statistics section does not mentions whether multiple testing corrections are applied or not. if so, I would expect the investigators to clarify which procedure was used.

Response: Thanks for your suggestion. For statistics, we used the LASSO method for variable selection, it is a high throughput method designed to circumvent the multiple testing when the number of variables are substantially larger than the number of samples. For LASSO method, there is no need for multiple test correction. UPLC-MS methods including chromatographic separation, mass spectrometry, analytical method assessment and data analysis have now been thoroughly described in the Supplementary Material in the revised manuscript, under the “**Analytical method assessment**” section.

Minor comments:

- Table S1-4 are redundant, it seems only 1 table is needed.

Response: Thanks for your kind suggestion, but the simultaneous remove of Table S1-4 may lose some information. Therefore, we have removed Table S1 and S2 without eliminating information, while Table S3 is now Table S1. Table S4 has been removed too, because the entire co-abundance analysis has been removed from the entire manuscript.

- The retention times and m/z of the metabolites should be listed.

Response: The retention times and m/z of the metabolites have been presented in new **Table S1**.

- Multivariate methods are mentioned but with little or no explanation on the parameters used.

Response: Overall statistical analyses were re-organized and significant details have been added to the Supplementary Materials, under the “**Model construction for variable selection and biomarker discovery**” section.

- Figures seem to be organised in a systematic way, unfortunately the files I have to review are not legible, which makes the assessment of barplots almost impossible

Response: Thanks for your kind suggestion. We have checked carefully and rearranged each figure to make it legible. The explanations of barplots have been added in animal and cell experiments.

Reviewer #3 - expert in metabolomics (Remarks to the Author):

In their manuscript “Metabolites strongly associated with estimated glomerular filtration rate and anti-fibrotic effect of 5-methoxytryptophan” the authors used a non-targeted metabolomics approach in serum from 703 subjects for the identification of potential biomarkers indicating early stages of chronic kidney disease (CKD). Five metabolites, including 5-methoxytryptophan (5-MTP) were highlighted and validated as biomarkers in independent cohorts and study designs (cross-sectional, longitudinal, pre-/post treatment). In addition, effects of modulating 5-MTP by supplementation or manipulation of TPH-1 expression were further evaluated using an animal model. Based on their analyses, the authors concluded that (i) the highlighted five metabolites are good biomarkers for early kidney disease (stages 1 and 2), (ii) 5-MTP is renoprotective, and iii) TPH-1 might be a promising drug target for treating CKD.

While several studies identified metabolites that allow early detection of kidney function decline by analyzing metabolomics data in large cohorts previously, I appreciate the depth of validation and experimental follow-up of the metabolites highlighted in the present work, which I would consider the study's major strength. Unfortunately, the approaches used for identifying the five highlighted metabolites are unclear and not described in sufficient detail and clarity in my opinion. Based on the provided information, I actually found it very difficult to judge the validity of the approach. Moreover, I would have expected a more thorough discussion of the results in the light of the related previous studies on the association of eGFR and metabolites in population-based cohorts, at least in the case of contradicting results, such as the opposite effect directions observed for tiglylcarnitine.

Please see below for more detailed comments:

1) Description of the stepwise feature selection using LASSO is unclear and needs much more details to allow judging the validity of the approach and its results:

Response: Thanks for your kind suggestion. The detail of feature selection using LASSO has been thoroughly extended in the Supplementary Material. Please see detail regarding each of the comments below.

a. The description of model construction is confusing and extremely spread in the supplement:

- The 8 lines of description provided in the Supplementary Material on top of pg. 1 under the title “Model construction for variable selection and biomarker discovery”, gives the impression that all 25,109 features went into the construction of the LASSO model (“In our study, we included 703 subjects in our initial dataset and 25109 features. By using LASSO regularization method, we obtained 98 metabolites from 25109 features.”). The very same text is repeated on pg. 4 of the supplement (“Model construction for variable selection and biomarker discovery”)!

- From Figure S1, and the paragraph entitled “UPLC-MS analysis” (?), I understand that you built separate models for positive mode features, negative mode features and clinical features (which ones?) in the first round of feature selection. The second round of feature selection is basically not described anywhere beyond Figure S1.

- On pg4., in the sections entitled “Model development, data analysis and differential metabolite identification”, there is one sentence informing the reader that LASSO was performed using the glmnet package and 10-fold cross validation. Unfortunately, results from the cross-validation are not provided anywhere.

- On pg. 5 of the supplement in the section “Biomarker selection and validation” (?) a short explanation for LASSO is provided stating how the LASSO method works in principle: “LASSO is a feature selection technique that uses least absolute shrinkage, ...”.

Response: The duplicated section has been removed from page 4 in Supplementary Material. The section “**Model construction for variable selection and biomarker discovery**” has been extensively rewritten to document the detail of the feature selection procedures. The scattered description of LASSO model has been consolidated into one section as well. After obtaining the 98 metabolites from both positive and negative ion modes, we felt they were overmuch for better clinical usage and functional validation follow up, so we needed to reduce the number of metabolite one more time. Thus we conducted one more round of LASSO on these 98 metabolites, which selected the final five metabolites. This has now been properly documented in the Supplementary Material, under the “**Model construction for variable selection and biomarker discovery**” section.

b. Some details that are important to allow following or repeating your approach are missing completely, for example: How did you select lambda in your two rounds of LASSO-based feature selection? How does the choice of lambda affect model fit? Which 17 clinical features were used?

Response: Our two rounds of LASSO-based feature selection were performed using the R package glmnet. The lambda values were chosen based on the Bayesian Information Criterion (BIC). In each round of feature selection, we fitted the

penalized ordinal response logistic regression with 100 candidate lambdas and calculated the BIC for each lambda. The optimal model was determined by the lambda corresponding to the minimal BIC. The choice of lambda based on the minimal BIC would fit the model very well without overfitting the data. All these procedures are implemented in the package glmnet. We have now added the detailed information about the model fitting including the choice of lambdas in the Supplementary Material. The 17 clinical variables were fitted with the same LASSO model, BUN and CREA were selected by the model. Furthermore, the 17 clinical variables were also used in the second round of the variable selection by LASSO with the 98 metabolites selected from the first round. We have made this clear in the Supplementary Material.

c. The results from the multivariate ordinal regressions in Tables S1, S2, and S3 are not very informative. In my opinion, information on the outcome of the LASSO models, in particular the final one, from which the five metabolites were taken, would be much more relevant. In addition, the effect sizes from univariate tests could be provided for each metabolite, which are especially interesting in the cases of highly correlated metabolites, for which the effect seems to be kind of “spread” over several compounds in the multivariate model. Some further questions/comments regarding Tables S1, S2, S3:

- Did you use any relevant covariates such as age, sex, BMI, and blood pressure, diabetes etc. in the models?
- Listing effects in a format so that most of them appear as 0.00 like in Table S3 is not very informative.
- Looking at Table S3, it is a bit surprising that none of the lipids (e.g. PC(33:3), DG(42:3), TG(68:7)) that showed low p-values in the ordinal regression was selected in the final LASSO. Is there any good explanation for that?

Response: The original Table S3 values were fitted using a multivariate model where all 98 metabolites were fitted simultaneously. Many of the 98 metabolites were highly correlated, causing confounding effects, and cancel out each other's effect, causing the p-value to be less significant. The new table (**Table S1**) now fits the 98 metabolite using univariate model as suggested, all of the metabolites are highly significant and displaying multiple decimal points now. No clinical variable were used in this univariate model, clinical variables were used in the LASSO variable selection.

d. My understanding is that metabolites measured in urine were not used for building the models. I would be curious to know why? In principle urine is more easily accessible than serum, which could be an advantage for actual clinical use.

Response: We thank the distinguished reviewer for raising this issue. We agree with the reviewer that the metabolites measured in the urine are eventually easier for clinical application. However, urine is a complex matrix containing a wide variety of acidic, neutral and basic compounds with high polarity. Various factors, such as age, sex, diet and urinary volume, have dramatic effects on the level of urinary metabolites. In addition, in the current manuscript, our study focuses on the serum samples rather than urinary samples by metabolomics. When we originally submit the manuscript to Nature Medicine,

we included targeted metabolite data on the five final metabolites from urinary samples. High throughput urine metabolite data is not available. We felt that the targeted urine metabolite data do not really add any significant finds to the manuscript, thus we have removed it from the manuscript. Thanks for your kind suggestion and we will focus on the urinary metabolomics in the future.

2) Information provided for the metabolomics approaches and the pre-processing of the data is extremely sparse. Though I agree that readers can be referred to previous publications for details about the methods, the analytical procedures and main steps in data processing, which are important for the interpretation of results, should at least be delineated in the Supplementary Methods in my opinion. Moreover, all the mentioned “minor modifications” (Suppl. Mat., pg. 3) from the published protocol should be clearly described. In addition:

Response: Thanks for your suggestion. UPLC-MS methods including chromatographic separation, mass spectrometry, analytical method assessment and data analysis including precision and repeatability have been described in the “**High throughput metabolomics and measurement of metabolites**” section of the Supplementary Material in the revised manuscript.

a. Please provide information on the level of confidence in your metabolite identification according to the guidelines of the Metabolomics Standards Initiative (MSI).

Response: The detail of metabolite identification is described now in the “**Metabolite Identification**” section of the Supplementary Material.

b. From Figure S2B it looks like missing values in the metabolomics data were set to zero or imputed with a low value. Maybe I have missed it, but I could not find any comment on the handling of missing values in the manuscript or the supplement.

Response: The raw data were processed by the commercial software Prognosis QI and Markerlynx XS developed by Nonlinear Dynamics, Waters Corporation. According to the software, missing value is considered undetectable, thus replaced as 0. We have updated this information in the method section in Supplementary Material, under the “**Analytical method assessment**” section.

3) Discussion:

a. While, in the present study, low levels of tiglycarnitine (C5:1) were found to associate with higher stages of CKD, various previous studies reported high levels of C5:1 to be linked to low eGFR [Goek et al., Am J Kidney Dis., 2012; PMID: 22464876. Sekula et al., J Am SocNephrol., 2016; PMID: 26449609], and diabetic kidney disease [Liu et al., Kidney Int Rep., 2017; PMID:29142974.]. Please discuss!

Response: Goek *et al.* reported high levels of tiglycarnitine (C5:1) to be linked to low eGFR based on KORA study in UK. Sekula *et al.* cited the Goek's study. Goek and Sekula' study are based on the KORA study in the UK. There are several possible interpretations. First, the several studies that included tiglycarnitine were all based on the same dataset from the KORA study. Furthermore, the metabolite data in the KORA study were collected from 100% Caucasians. Our study was different from the KORA study. The diets and ethnic backgrounds of the study populations may play a role in the difference. We have discussed this result in the revised manuscript. There have been few studies of tiglycarnitine in CKD, Maeda *et al* indicated no detected tiglycarnitine in serum and the decreased urine tiglycarnitine level (1 year) and increased urine tiglycarnitine level (3 year) in patient with multiple carboxylase deficiency (J Chromatogr B 2008;870:154-159). Fukao *et al* indicated the urinary tiglyglycine was not or was only faintly detected in patients with mitochondrial acetoacetyl-CoA thiolase deficiency (J Inherit Metab Dis 2003; 26: 423-431). Fontaine *et al* reported increased tiglycarnitine level in patients with 2-methylacetoacetyl-CoA thiolase deficiency (Clin Chim Acta 1996;255:67-83).

Based on targeted plasma metabolomics, Liu *et al.* identified 123 plasma metabolites in patients with early DKD and overt DKD and further validated the identified metabolites in patients with macroalbuminuria and matched non-DKD T2DM controls. 123 Plasma metabolites including 16 amino acids, 9 short-chain acylcarnitines, 12 medium-chain acylcarnitines, 24 long-chain acylcarnitines, 7 krebs cycle organic acids, 25 ceramides, 12 sphingomyelin, sphinganine and sphingosine and, 19 phosphatidylcholines were absolutely quantified. We wanted to compare our results with Liu's study, however, metabolomic method in Liu's study only referenced other publications such as Muoio *et al*, Cell Metab 2009;9:311–326; Sinha *et al*, Hepatology 2014;59:1366–1380 and Loh *et al*, Nephrology 2015;20:216–223. In Supplementary Methodology, the paper only refers the names of liquid chromatography and mass spectrometry but did not mention the experimental condition or apparatus parameters in detail for LC-MS. This publication did not provide the complete experimental materials and methods. Since the information provided by Liu's study was incomplete, it is difficult for us to compare our study with Liu's.

b. Pg 7, second paragraph: acylcarnitine (i.e., the sum of molecules with a fatty acid (various lengths and desaturation) linked to carnitine) and acetylcarnitine (C2) appear to be used like synonyms in this paragraph. It was not always clear to me to which of the two you actually wanted to refer.

Response: Sorry about this confusion. We meant to say acetylcarnitine. We have changed the “acylcarnitine to acetylcarnitine”.

4) Validation models:

a. When analyzing the data from the drug treatment cohort, wouldn't it be more interesting to examine how changes in the five metabolites correlate with the improvement of proteinuria instead of “predicting” pre-/post treatment?

Response: Thanks for your question. Although proteinuria was an important parameter for diagnosis and prediction of drug therapeutic effects on CKD, we aimed to find alternative biomarkers for the prediction of therapeutic effects of drugs in pre- and post treatment. Our results demonstrated that five metabolites could be considered as alternative biomarkers for the prediction of therapeutic effects of drugs.

b. Pg 4: Figure 3K does not exist. Did you mean 3G?

Response: Thanks for your kind reminder. It should be Figure 3G. We have corrected this in the text.

5) Co-abundance/correlation analysis:

a. Metabolites are typically highly correlated even if they are only indirectly connected. Therefore, co-abundance networks based on metabolomics data usually do not reflect metabolic pathways well.

Response: We agree with review on this point. Also in response to part c of this comment, we have removed co-abundance and correlation analyses from the manuscript.

b. Figure S3A/Table S4: In the Figure caption (and table header), you wrote that you analyzed correlation between the five “potential biomarkers and 337 metabolites”. However, my understanding was that you only identified metabolites represented by the selected features (98). Do you mean 337 metabolite pairs with correlation >0.9 (assuming that “P value” in the header of Table S3 is a mistake)? The caption also reads as if all correlations > 0.9 are shown in Figure S3A. If so, why is acetylcarnitine only linked to one metabolite in the figure but has much more correlations >0.9 according to Table S4?

Response: Sorry for the confusion. The 98 metabolites were selected by the first round of LASSO regression. The second round of LASSO regression selected five metabolites from the 98 metabolites. Then we identified 337 metabolites that were correlated with these five metabolites. The reviewer is correct that the header “P value” in the original Table S3 is wrong. We are sorry that we mislabeled tiglylcarnitine and acetylcarnitine. It was tiglylcarnitine that had only 1 correlation > 0.9 .

c. In general, I cannot see how the analysis of co-abundance/correlation of the measured metabolites or the pathway enrichment analysis adds to or supports the claim of the paper. Results are only mentioned with one sentence in the manuscript (pg. 3: “Pathway and co-abundant analyses shed light on potential metabolic process of CKD (Supplementary Material and Figure S2, S3)”). Maybe I missed it, but I could not find any description or discussion related to these “potential metabolic processes”. I would suggest to either describe them explicitly in the Results and Discussion sections or to remove the analysis completely from the manuscript.

Response: Thanks for your suggestion. To make the manuscript clearer and more accessible, we have removed the co-abundance and correlation analyses from the manuscript.

6) Is there any rational behind focusing on TPH-1 but not on other genes in the pathway of 5-MTP such as ASMT2 or IDO2?

Response: Thanks for your question. IDO2 usually mediates in the catabolism of tryptophan to kynurenine (Clin Immunol. 2017 Jun;179:8-16; Int J Tryptophan Res. 2017 Oct 9;10:1178646917735098), while ASMT2, also known as HIOMT, is the final enzyme in a biosynthetic pathway that produces melatonin, another important metabolite of tryptophan (J Pineal Res. 2013 Nov;55(4):409-415;). As you said, 5-MTP can be produced by several enzymes, TPH-1 and ASMT2 included. However, among several enzymes, TPH-1 is more attractive than others. Several studies demonstrated that the modulation of 5-MTP by TPH-1 show a regulatory role in many diseases, including lung cancer (Proc Natl Acad Sci U S A 2012;109(33):13231-13236; Oncotarget 2016;7(21):31243-31256), inflammation in human foreskin fibroblasts (PLoS One 2014;9(2):e88507) and endothelial impairment in human umbilical vein endothelial cells (Circ Res. 2016 Jul 8;119(2):222-36). Therefore, TPH-1 was selected to regulate 5-MTP expression in the present study. TPH-1 enables conversion of *L*-tryptophan to 5-MTP, which was a focus of our previous study. In the present study, we found that TPH-1 could regulate 5-MTP level and the regulation of 5-MTP by TPH-1 significantly affected the progression of renal disease.

7) Typo on pg. 7, first line: THP-1 should be TPH-1.

Response: Thank you for pointing this out, we have corrected this.

Reviewers' comments:

Reviewer #1 (Remarks to the Author):

The authors have addressed my major concerns. A few minor questions remain:

1. TPH-1 immunostaining. Is it restricted to renal tubular cells? The image for CKD4 looks like sporadic interstitial cells are also positive but the resolution is too low to be certain.
2. A bit more discussion is needed to understand how the authors believe that the enalapril data should be interpreted. Given that the focus of the metabolomics studies was CKD stage defined by eGFR and not proteinuria levels and eGFR did not change significantly during the 6-month treatment study, what do higher 5-MTP levels mean in this context?

Reviewer #2 (Remarks to the Author):

In this revised version, the authors present a refined manuscript on the role of 5-methoxytryptophan (5MTP) as a marker of chronic kidney disease with antifibrotic properties.

Whilst the authors have made progress and clarified certain sections of the manuscript and simplified it, I still have major concerns about this manuscript as the authors have not fully addressed some of the comments I had made.

In particular, there are many assumptions made by the investigators when choosing machine learning algorithms such as support vector machines, random forests and the lasso. These models are not covariate- or confounder-adjusted and this is one of the main drawbacks of the data analysis strategy. The presence of replication makes the results stable but these could still be confounded by risk factors and other biases. Using a 2-step variable selection with the LASSO is prone to overfitting. Along the same lines, the variable selection(s) and the classification model should be validated at the same time during the crossvalidation (that is, performing a new variable selection and building a calibration model for each calibration set).

The metabolite lists and therefore the biological interpretations in relation to CKD could still be skewed. For instance, 5MTP has anti-inflammatory and anti-fibrotic activities and overexpression of

its biosynthetic enzyme TPH-1 reduces renal injury. I could have missed it but 5MTP does not seem to directly improve renal function.

Reviewer #3 (Remarks to the Author):

While I appreciate that the authors have addressed many of my comments in the revised manuscript, there are still some issues that have not been resolved in my opinion:

Regarding my original comments under 1)

- I understand that the urine data have been removed from the manuscript. Though I probably wouldn't have missed them if they had been absent from the very beginning, these data showed that 5-MPH increases in urine in higher stages of CKD (see also comment by Reviewer #1: "THP-1 levels appear to decline with renal injury yet urinary 5-MTP levels (Figure 1) increase. How is this explained?"). This raises the question whether it is really the production of 5-MTP that is going down with the decline of kidney function as (at least implicitly) assumed in the manuscript. In my opinion, this finding should not be ignored in the manuscript by just removing the urine data. Rather this should be briefly discussed.

- Unfortunately, the authors did not comment on my following question from my original review: "Looking at Table S3, it is a bit surprising that none of the lipids (e.g. PC(33:3), DG(42:3), TG(68:7)) that showed low p-values in the ordinal regression was selected in the final LASSO. Is there any good explanation for that?". In my opinion, it would be worth to very briefly discuss possible explanations.

Regarding my original comments under 2)

- While I appreciate that the authors added a thorough description of how they do metabolite identification in principle, the manuscript still lacks information on the actual level of identification for each identified metabolite according to the guidelines of the Metabolomics Standards Initiative (MSI). For which metabolites did the authors run the respective pure substance on their own platform (identification level 1) and which metabolites were identified only by comparison with spectra from literature or public databases (level 2 or 3)? This information should be added to Supplementary Table S1.

- In the section "UPLC-HDMS method assessment" in the supplement, the statement "The repeatability and precision were tested by six reduplicate analyses from the quality control samples and six samples, respectively. Relative standard deviation (RSD%) of retention time and peak area were below 0.52% and 2.9%, respectively. This method showed the good repeatability and precision in this study." is written three times (with slight variations) on half a page.

Regarding my original comments under 3)

I appreciate that the authors now mention that higher tiglylcarnitine (C5:1) levels have been found to associate with lower eGFR and CKD in previous studies [e.g., eGFR: Goek et al., *Am J Kidney Dis.*, 2012; PMID: 22464876. Sekula et al., *J Am SocNephrol.*, 2016; PMID: 26449609; diabetic kidney disease: Liu et al., *Kidney Int Rep.*, 2017; PMID:29142974], which is in contrast to the findings in the present study. Differences in reported effect directions indeed might be explained by differences in the ethnicity or diet of the underlying cohorts. However, ethnic differences do not explain why Liu et al., which - same as the present study - was based on Asians, also found higher tiglylcarnitine to occur in kidney disease. Unfortunately, the authors chose not to discuss the latter study at all in the revised manuscript. While I agree that experimental procedures should always be provided in sufficient detail, which might not be the case in Liu et al. (I did not check this), it is not necessary to repeat the measurements as they were done in Liu et al. to discuss the differences in findings, in my opinion. Of note, while it is correct that Goek et al. and Sekula et al. were based on Caucasians using the same cohort for discovery but applying two different metabolomics approaches in the two studies, KORA is not a UK cohort as wrongly stated in the rebuttal. In fact, KORA is a German cohort and findings in KORA were replicated in an independent UK cohort in both studies.

Responses to reviewers' comments NCOMMS-18-11249A

We thank the reviewers for their careful review of our manuscript, and constructive comments, which have guided our revision of our work (Manuscript ID NCOMMS-18-11249A). The changes in the manuscript have been highlighted in yellow for easy identification. The concerns have been addressed as follows:

Reviewers' comments:

Reviewer #1 (Remarks to the Author):

The authors have addressed my major concerns. A few minor questions remain:

1. TPH-1 immunostaining. Is restricted to renal tubular cells? The image for CKD4 looks like sporadic interstitial cells are also positive but the resolution is too low to be certain.

Response: We agree with the reviewer and are submitting a higher resolution images. Renal injury results in the downregulation of TPH-1, which is expressed in kidney tissue but is not restricted to renal tubular cells.

CKD4

2. A bit more discussion is needed to understand how the authors believe that the enalapril data should be interpreted. Given that the focus of the metabolomics studies was CKD stage defined by eGFR and not proteinuria levels and eGFR did not change significantly during the 6-month treatment study, what do higher 5-MTP levels mean in this context?

Response: Thanks for your question. During the 6-month treatment study, eGFR did not change significantly but proteinuria was significantly decreased, indicating the therapeutic benefit of enalapril. In the present study, five metabolites were selected to identify healthy controls from patients with five stages of CKD; they performed well while eGFR is not enough sensitive in early stages of CKD, indicating higher sensitivity of the five metabolites than for changes in eGFR. A significant increase in 5-MTP was observed but eGFR did not change significantly, which may result from the lower sensitivity of eGFR. In addition, we have discussed these in the revised manuscript. All authors appreciated your hard work that helped us improve the manuscript.

Reviewer #2 (Remarks to the Author):

In this revised version, the authors present a refined manuscript on the role of 5-methoxytryptophan (5MTP) as a marker of chronic kidney disease with antifibrotic properties.

Whilst the authors have made progress and clarified certain sections of the manuscript and simplified it, I still have major concerns about this manuscript as the authors have not fully addressed some of the comments I had made.

In particular, there are many assumptions made by the investigators when choosing machine learning algorithms such as support vector machines, random forests and the lasso. These models are not covariate- or confounder-adjusted and this is one of the main drawbacks of the data analysis strategy. The presence of replication makes the results stable but these could still be confounded by risk factors and other biases. Using a 2-step variable selection with the LASSO is prone to overfitting. Along the same lines, the variable selection(s) and the classification model should be validated at the same time during the crossvalidation (that is, performing a new variable selection and building a calibration model for each calibration set).

Response: Regarding the machine learning methods used, only LASSO was used as feature selection method for detecting metabolite biomarkers. The other machine learning methods were used for validation purpose. While using LASSO for feature selection, we did adjust for clinical variables. The final five metabolites were selected by LASSO with clinical variables in the model, this should have considered the confounding effect. Furthermore, the goal of the current study is to identify novel metabolite biomarkers for early stage CKD. The established clinical biomarkers such CREA and BUN can distinguish later stage CKD but not early stage CKD from normal subjects (Figure 1). Any remaining confounding effect between metabolites and clinical variable is not the focus of the study.

Furthermore, we re-performed sevenfold cross validation to obtain an unbiased estimate for the model performance. We randomly divided the dataset into 10 partitions. In each fold of cross-validation we took one part as a test set and combined the rest of nine parts as a training set. We repeated our two-step LASSO-based feature selection with the training set and obtained a prediction model. Then we used this model to prediction on the test set and calculate a prediction accuracy. The final accuracy is an average of all the accuracies obtained from all the folds of the cross validation, which is 97.2%. This information has been updated in the supplementary material.

The metabolite lists and therefore the biological interpretations in relation to CKD could still be skewed. For instance, 5MTP has anti-inflammatory and anti-fibrotic activities and overexpression of its biosynthetic enzyme TPH-1 reduces renal injury. I could have missed it but 5MTP does not seem to directly improve renal function.

Response: Thanks for your question. As shown in Figure 5 and 6, the anti-inflammatory and anti-fibrotic activities of 5-MTP were explored *in vivo* and *in vitro*, respectively. In Figure 5, 5-MTP treatment inhibited NF- κ B pathway, activated Nrf2 pathway and attenuated fibrosis in UUO mice. In Figure 6, 5-MTP treatment decreased LPS-induced inflammation in HMC cells and alleviated TGF- β 1-induced fibrosis in HK-2 cells.

Reviewer #3 (Remarks to the Author):

While I appreciate that the authors have addressed many of my comments in the revised manuscript, there are still some issues that have not been resolved in my opinion:

Regarding my original comments under 1)

- I understand that the urine data have been removed from the manuscript. Though I probably wouldn't have missed them if they had been absent from the very beginning, these data showed that 5-MTP increases in urine in higher stages of CKD (see also comment by Reviewer #1: "THP-1 levels appear to decline with renal injury yet urinary 5-MTP levels (Figure 1) increase. How is this explained?"). This raises the question whether it is really the production of 5-MTP that is going down with the decline of kidney function as (at least implicitly) assumed in the manuscript. In my opinion, this finding should not be ignored in the manuscript by just removing the urine data. Rather this should be briefly discussed.

Response: We thank the distinguished reviewer for careful reviewing our manuscript. In the present study, decreased 5-MTP in serum and increased 5-MTP in urine were observed during CKD progression. As far as we know, the level of 5-MTP was hardly reported in urine of patients with CKD or CKD animal models. Hence, it was difficult to directly discuss the reason that caused the increase of 5-MTP in urine. Interestingly, we found that the trends in 5-MTP and tryptophan were similar during CKD progression in that both 5-MTP and tryptophan were decreased in serum and

increased in urine. Since it has been reported that 5-MTP was synthesized from *L*-tryptophan via tryptophan hydroxylase-1 (TPH-1) that was an important resource for 5-MTP. (Cheng *et al*, Proc Natl Acad Sci USA 2012;109:13231-13236; Cheng *et al*, Oncotarget 2016;7:31243-31256; Wang *et al*, Circ Res 2016;119:222-236), we supposed that decreased serum tryptophan and increased urinary tryptophan were one of the reason for decreased serum 5-MTP and increased urinary 5-MTP. For the level of tryptophan in serum and urine of patients with CKD or CKD animal model, we discussed as follows:

Several studies have reported the decreased tryptophan in serum and increased tryptophan in urine from patients with different stages of CKD (Duranton *et al*, Clin J Am Soc Nephrol 2014;9:37-45; Bao *et al*, Biomarkers 2013;18:379-385; Chen *et al*, Redox Biol 2017;12:505-521). Similar results (decreased tryptophan in serum and increase tryptophan in urine) were also observed in the adenine-CRF rats compared with control rats (Zhao *et al*, Biomarkers 2012;17:48-55; Zhao *et al*, Clin Chim Acta 2012;413:642-649). In addition, compared with healthy controls, other studies also demonstrated the decrease in serum tryptophan from patients with CKD (Zhao *et al*. Ren Fail 2013;35:648-653; Toyohara *et al*, Hypertens Res 2010;33:944-952), patients with CRF (Jia *et al*, Metabolomics 2008;4:183–189; Pawlak *et al*, Thromb Res 2009;124:452-457), patients with ESRD (Rhee *et al*, J Am Soc Nephrol 2010;21:1041-1051) and uremic patients (Karu *et al*, BMC Nephrol 2016;17:171). Increase in urinary tryptophan was observed in patients with diabetic kidney disease (Van der Kloet *et al*, Metabolomics 2012;8:109-119).

For animal models, it has also been reported that decreased plasma concentration and increased urinary excretion of tryptophan was observed in mice with chemically-induced kidney injury (Zgoda-Pols *et al*. Toxicol Appl Pharmacol 2011;255:48-56) and PDK1 hypomorphic mice (Rexhepaj *et al*, FASEB J 2006;20:2214-2222). In addition, several studies demonstrated the decrease in serum tryptophan from control rats to mild CKD and severe CKD rats (Saito *et al*, Am J Physiol Renal Physiol 2000;279:F565-F572; Pawlak *et al*, J Physiol Pharmacol 2003 Jun;54:175-189; Pawlak *et al*, J Physiol Pharmacol 2001;52:755-766; Pawlak *et al*, Nephron 2002;90:328-335; Kalaska *et al*. PeerJ 2017;5:e3199). Moreover, Ganti *et al* demonstrated that changes in metabolites were variously concordant and discordant in tissue, serum and urine in mice with kidney cancer. Levels of 68 metabolites in serum and urine in the same direction changed concordantly, while 28 metabolites in serum and urine changed discordantly. Tryptophan level were decreased in kidney tissues and serum while its level was increased in urine (Ganti *et al*, Cancer Res 2012;72:3471-2479).

It has been certain that 5-MTP level was closely associated with tryptophan level (Cheng *et al*, Proc Natl Acad Sci U S A 2012; Wang *et al*, Circ Res. 2016;119:222-36), and the similar trends in 5-MTP and tryptophan levels in serum and urine were observed; hence, we deduced that increased urinary tryptophan contributed to increased urinary 5-MTP.

Secondly, reduced renal function could significantly affect the serum and urine concentrations of many metabolites in

CKD. Zhao *et al* reported that impaired tubular reabsorption function caused decreased plasma tryptophan concentration in CKD (Zhao *et al*. *Ren Fail* 2013;35:648-653). This might be one reason of increased urinary tryptophan, and further contributed to increased urinary 5-MTP. Increased 5-MTP in urine was partly associated with impaired tubular reabsorption function.

- Unfortunately, the authors did not comment on my following question from my original review: “Looking at Table S3, it is a bit surprising that none of the lipids (e.g. PC(33:3), DG(42:3), TG(68:7)) that showed low p-values in the ordinal regression was selected in the final LASSO. Is there any good explanation for that?”. In my opinion, it would be worth to very briefly discuss possible explanations. Regarding my original comments under 2) - While I appreciate that the authors added a thorough description of how they do metabolite identification in principle, the manuscript still lacks information on the actual level of identification for each identified metabolite according to the guidelines of the Metabolomics Standards Initiative (MSI). For which metabolites did the authors run the respective pure substance on their own platform (identification level 1) and which metabolites were identified only by comparison with spectra from literature or public databases (level 2 or 3)? This information should be added to Supplementary Table S1.

Response: The results from the univariable analyses and from the lasso variable selection can be inconsistent. A variable/feature may show very strong effect on the outcome in the univariable analysis but not selected by the lasso, and vice versa. However, this is anticipated. Univariable analysis is the simplest form of statistical analysis which involves only one variable. The significance of the relationship between any given variable with the outcome is based solely on that variable and does not consider the presence of other variables. While in reality, it is always important not only to evaluate the effect of that variable alone, but also in combination with other variables (in our situation, clinical factors and other metabolites), some of which may be correlated. It is common to see a variable shows significance in univariable analysis but insignificant in multivariable analysis, after adjusting for other variables in the model. In lasso type of analysis, in particular, when a number of highly correlated variables are associated with the outcome, lasso tends to only select one of them. Therefore, it is possible that is correlated with a selected feature. With the presence of the selected feature, PC(33:3), DG(42:3), TG(68:7) were not needed in the final model, while this cluster of correlated variables are actually equally important (this is also for parsimony). We have updated discussion to explain this point.

The actual level of identification for each identified metabolite was presented in Table S1 according to the guidelines of the Metabolomics Standards Initiative (MSI). Table S1 showed that the metabolites were identified by the respective pure substance or only comparison with spectra from literature or public databases. “&” indicated the metabolites identified by the respective pure substance.

- In the section “UPLC-HDMS method assessment” in the supplement, the statement “The repeatability and precision were tested by six reduplicate analyses from the quality control samples and six samples, respectively. Relative standard deviation (RSD%) of retention time and peak area were below 0.52% and 2.9%, respectively. This method showed the good repeatability and precision in this study.” is written three times (with slight variations) on half a page.

Response: Thanks for your reminder. The duplicate description has been deleted.

Regarding my original comments under 3)

I appreciate that the authors now mention that higher tiglylcarnitine (C5:1) levels have been found to associate with lower eGFR and CKD in previous studies [e.g., eGFR: Goek et al., *Am J Kidney Dis.*, 2012; PMID: 22464876. Sekula et al., *J Am SocNephrol.*, 2016; PMID: 26449609; diabetic kidney disease: Liu et al., *Kidney Int Rep.*, 2017; PMID:29142974], which is in contrast to the findings in the present study. Differences in reported effect directions indeed might be explained by differences in the ethnicity or diet of the underlying cohorts. However, ethnic differences do not explain why Liu et al., which - same as the present study - was based on Asians, also found higher tiglylcarnitine to occur in kidney disease. Unfortunately, the authors chose not to discuss the latter study at all in the revised manuscript. While I agree that experimental procedures should always be provided in sufficient detail, which might not be the case in Liu et al. (I did not check this), it is not necessary to repeat the measurements as they were done in Liu et al. to discuss the differences in findings, in my opinion. Of note, while it is correct that Goek et al. and Sekula et al. were based on Caucasians using the same cohort for discovery but applying two different metabolomics approaches in the two studies, KORA is not a UK cohort as wrongly stated in the rebuttal. In fact, KORA is a German cohort and findings in KORA were replicated in an independent UK cohort in both studies.

Response: Thank you very much for your suggestion. We agree with your comments about Liu *et al*, and have added the discussion in the revised manuscript. We have corrected our mistake in identifying KORA as a UK cohort and revised the manuscript to identify it as a German cohort. All authors appreciated your hard work that helped us improve the manuscript.

REVIEWERS' COMMENTS:

Reviewer #1 (Remarks to the Author):

I am satisfied with the responses to my questions.

Reviewer #3 (Remarks to the Author):

The authors have addressed most of my major concerns in the revised manuscript. However, in one case they chose to respond without adapting the manuscript in any way. I would like to leave it for the editors' discretion whether this point has to be added in the manuscript or not:

I appreciate the thorough literature search on associations of 5-MPH in urine versus blood which the authors described as a response to my comment. However, the authors did not include any notice on the 5-MPH levels being higher in urine with lower kidney function in contrast to blood levels. Maybe I was not clear about my main point here: I am not questioning the value of targeting THP-1, as enough evidence has been provided in the manuscript for that, I think. The only point that I tried to make is that the findings in urine should not be ignored and left undiscussed. The increased levels of 5-MPH in urine suggest that the lower levels of 5-MPH in blood might not necessarily be due to lower production of 5-MPH from tryptophan (as at least implicitly assumed in the manuscript) but due to reduced reabsorption of tryptophan and 5-MPH in the kidney. In my opinion, it would be worth to mention this point with one or two sentences in the discussion.

Further comment:

Pg. 7, last paragraph: "Acetylcarnitine is a carrier of long-chain fatty acids through the mitochondrial inner membrane for β -oxidation." does not sound correct to me. Did you mean "acylcarnitines" in general rather than "acetylcarnitine", which is a short-chain (C2) acylcarnitine?

Responses to reviewers' comments NCOMMS-18-11249B

We thank the editor and reviewers for their careful review of our manuscript, and constructive comments, which have guided our revision of our work (Manuscript ID NCOMMS-18-11249B). The changes in the manuscript have been highlighted in yellow for easy identification. The concerns have been addressed as follows:

REVIEWERS' COMMENTS:

Reviewer #1 (Remarks to the Author):

I am satisfied with the responses to my questions.

Response: We thank again the distinguished reviewer for careful reviewing our manuscript.

Reviewer #3 (Remarks to the Author):

The authors have addressed most of my major concerns in the revised manuscript. However, in one case they chose to respond without adapting the manuscript in any way. I would like to leave it for the editors' discretion whether this point has to be added in the manuscript or not:

I appreciate the thorough literature search on associations of 5-MPH in urine versus blood which the authors described as a response to my comment. However, the authors did not include any notice on the 5-MPH levels being higher in urine with lower kidney function in contrast to blood levels. Maybe I was not clear about my main point here: I am not questioning the value of targeting THP-1, as enough evidence has been provided in the manuscript for that, I think. The only point that I tried to make is that the findings in urine should not be ignored and left undiscussed. The increased levels of 5-MPH in urine suggest that the lower levels of 5-MPH in blood might not necessarily be due to lower production of 5-MPH from tryptophan (as at least implicitly assumed in the manuscript) but due to reduced reabsorption of tryptophan and 5-MPH in the kidney. In my opinion, it would be worth to mention this point with one or two sentences in the discussion.

Response: We thank the distinguished reviewer for careful reviewing our manuscript. We agree your comments that “the findings in urine should not be ignored and left undiscussed”, so we have added urine metabolomic results in Figure 1 and briefly discussed in the manuscript.

Further comment:

Pg. 7, last paragraph: “Acetylcarnitine is a carrier of long-chain fatty acids through the mitochondrial inner membrane for

β -oxidation.“ does not sound correct to me. Did you mean “acylcarnitines” in general rather than “acetylcarnitine”, which is a short-chain (C2) acylcarnitine?

Response: Thank you for your question. Acetylcarnitine is an acetic acid ester of carnitine that facilitates movement of acetyl-CoA into the matrices of mammalian mitochondria during the oxidation of fatty acids. The statement has been rewritten in the revised manuscript.